# Estimating degree-day factors based on energy flux components

**Muhammad Fraz Ismail [1, 2], Wolfgang Bogacki [2], Markus Disse [1], Michael Schäfer [2,3] and Lothar Kirschbauer [2]**

[1]  TUM School of Engineering and Design, Technical University of Munich, Germany

[2]  Department of Civil Engineering, Koblenz University of Applied Sciences, Germany

[3]  Faculty of Agriculture, Yamagata University, Tsuruoka, Japan

**Correspondence:** Muhammad Fraz Ismail (fraz.ismail@tum.de; ismail@hs-koblenz.de)

## Abstract

Meltwater from mountainous catchments dominated by snow and ice is a valuable source of fresh water in many regions. At mid-latitudes, seasonal snow cover and glaciers act like a natural reservoir by storing precipitation during winter and releasing it in spring and summer. Snowmelt is usually modelled either by energy balance or by temperature-index approaches. The energy balance approach is process-based and more sophisticated but requires extensive input data, while the temperature-index approach uses the degree-day factor (*DDF*) as key parameter to estimate melt merely from air temperature. Despite its simplicity, the temperature-index approach has proved to be a powerful tool for simulating the melt process especially in large and data scarce catchments.

The present study attempts to quantify the effects of spatial, temporal, and climatic conditions on the *DDF*, in order to gain a better understanding which influencing factors are decisive under which conditions. The analysis bases on the individual energy flux components, however formulas for estimating the *DDF* are presented to account for situations where observed data is limited. A detailed comparison between field-derived and estimated *DDF* values yields a fair agreement with bias = 0.14 mm °C$^{-1}$ d$^{-1}$ and Root Mean Square Error (RMSE) =1.12 mm °C$^{-1}$ d$^{-1}$.

The analysis of the energy balance processes controlling snowmelt indicates that cloud cover and snow albedo under clear sky are the most decisive factors for estimating the *DDF*. The results of this study further underline that the *DDF* changes as the melt season progresses and thus also with altitude, since melting conditions arrive later at higher elevations. A brief analysis of the *DDF* under the influence of climate change shows that the *DDFs* are expected to decrease when comparing periods of similar degree-days, as melt will occur earlier in the year when solar radiation is lower and albedo is then likely to be higher. Therefore, the *DDF* cannot be treated as a constant parameter especially when using temperature-index models for forecasting present or predicting future water availability.

**Keywords:** Degree-Day Factor, Snowmelt, Energy balance, Temperature-Index, Climate change

## 1. Introduction

Meltwater from snow and ice dominated mountainous basins is a main source of fresh water in many regions. Seasonal snow cover and glaciers act as natural reservoirs which significantly affect catchment hydrology by temporarily storing and releasing water on various time scales (Jansson et al., 2003). In such river basins, snow and glacier melt runoff modelling is a valuable tool when predicting downstream river flow regimes, as well as when assessing the changes in the cryosphere associated with climate change (Hock, 2003). Therefore, a more accurate quantification of the melt processes and related parameters is the key to a successful runoff modelling of present and future water availability.

Two different approaches are common in snowmelt modelling. The energy balance approach is process-based but data-intensive, since melt is deduced from the balance of in- and outgoing energy components (Braithwaite, 1995a; Arendt and Sharp, 1999). On the contrary, temperature-index or also-called degree-day models merely use the air temperature as an index to assess melt rates (Martinec, 1975; Bergström, 1976; Quick and Pipes, 1977; DeWalle and Rango, 2008). The degree-day approach is very common and popular since air temperature is an excellent surrogate variable for the energy available in near-surface atmosphere that governs the snowmelt process (Lang and Braun, 1990). The relationship between temperature and melt is defined by the degree-day factor ($DDF$) (Zingg, 1951; Braithwaite, 2008), which is the amount of melt that occurs per unit positive degree-day (Braithwaite, 1995a; Kayastha et al., 2003; Martinec et al., 2008). There are different methods by which the $DDF$ can be determined, e.g. by measurements using ablation stakes (Zhang et al., 2006), using snow lysimetric outflows (Kustas et al., 1994), by estimating daily changes in the snow water equivalent (Martinec, 1960; Rango and Martinec, 1979, 1995; Kane et al., 1997), or using satellite based snow cover data (Asaoka and Kominami, 2013; He et al., 2014).

The $DDF$ is usually treated as a decisive parameter subject to model calibrations because sufficient direct observations are typically lacking in large catchments. Most commonly, for calibrating the $DDF$, runoff is used (Hinzman and Kane, 1991; Klok et al., 2001; Luo et al., 2013; Bogacki and Ismail, 2016). However, it is also important to note that the calibration of the $DDF$ using runoff can be significantly affected by other model parameters due to their interdependency (Gafurov, 2010; He et al., 2014). Researchers also select $DDF$ directly from other studies, hence the spatial transferability is not always good (e.g. Carenzo et al., 2009; Wheler, 2009). Despite its simplicity, this approach has proved to be a powerful tool for simulating the complex melt processes especially in large and data scarce catchments (Zhang et al., 2006; Immerzeel et al., 2009; Tahir et al., 2011; Lutz et al., 2016).

Extensive research has been devoted to the enhancement of the original degree-day approach. Braun, (1984) introduced the Temperature-Wind-Index method by the inclusion of a wind-dependent scaling factor. A hybrid approach, which combines both, temperature-index and energy balance methods was introduced by Anderson, (1973). Hock, (1999) attempted to improve the simple temperature-index model by adding a term to consider potential incoming direct solar radiation for clear sky conditions.

The potential clear sky solar radiation is calculated as a function of the position of the sun, geographic location and a constant atmospheric transmissivity (Hock and Noetzli, 1997; Hock, 1999). This model is comparable with the data requirements of a simple degree-day model. Pellicciotti et al., (2005), considered the net shortwave radiation instead of just incoming shortwave radiation by including snow albedo in their proposed degree-day model. Although all these enhancements focus on adding more physical foundation to the original degree-day method, the classical approach is still more popular because of its simplicity and merely dependence on air temperatures.

A weakness of the degree-day approach is the fact that it works well over longer time periods (e.g. 10-daily, monthly, seasonal) but with increasing temporal resolution, in particular for sub-daily time-steps, the accuracy decreases (Lang, 1986; Hock, 1999). In addition, the spatial variability of melt rates is not modelled accurately as the *DDFs* are usually considered invariant in space. However, melt rates can be subject to substantial small-scale variations, particularly in high mountain regions due to topography (Hock, 1999). For example, topographic features (e.g. topographic shading, aspect and slope angles) including altitude of a basin can influence the spatial energy conditions for snowmelt and lead to significant variations of the *DDF* (Hock, 2003; Marsh et al., 2012; Bormann et al., 2014). Under otherwise similar conditions, *DDFs* are expected to increase with (i) increasing elevation, (ii) increasing direct solar radiation input and (iii) decreasing albedo (Hock, 2003).

Obviously, the *DDF* cannot be treated as a constant parameter as it varies due to the changes in the physical properties of the snowpack over the snowmelt season (Rango and Martinec, 1995; Prasad and Roy, 2005; Shea et al., 2009; Martinec et al., 2008; Ismail et al., 2015; Kayastha and Kayastha, 2020). The spatio-temporal variation in the *DDF* (Zhang et al., 2006; Asaoka and Kominami, 2013) not only affects the accuracy of snow and ice melt modelling (Quick and Pipes, 1977; Braun et al., 1993; Schreider et al., 1997) but also is a key to estimate heterogeneity of the snowmelt regime (Hock, 1999, 2003; DeWalle and Rango, 2008; Braithwaite, 2008; Schmid et al., 2012). Since melt depends on energy balance processes and topographic settings, changes in *DDFs* are a result of energy components that vary with different climatic conditions (Ambach, 1985; Braithwaite, 1995a). Another topic that needs attention is the stationarity of the *DDF* under climate change (Matthews and Hodgkins, 2016). Future water availability under climate change scenarios is typically modelled with *DDFs* calibrated for the present climate, which increases the parametric uncertainty introduced by the hydrological models (Lutz et al., 2016; Ismail and Bogacki, 2018; Hasson et al., 2019; Ismail et al., 2020).

In order to allow for a more process-based estimate of the *DDF*, present study attempts to quantify the contribution of each energy balance component to melt and subsequently to the overall *DDF*. Considering that degree-day models are typically utilised in large catchments with data scarce conditions, energy balance components are estimated by formulas with minimum data requirement following the approach by Walter et al., (2005). Based on these formulas, the *DDF* contribution corresponding to the respective energy components is quantified in tables and graphs for common snowmelt conditions, which can be used for a rapid appraisal. The presented approach is open in the

sense that if for any of the energy balance components observed data is available or more sophisticated models are desired, these can easily replace each of the presented approximations.

It shall be emphasised, that the objective of this study is not to incorporate an energy balance based *DDF* approach into temperature-index models. The aim is rather to gain a quantitative idea how different factors affect the *DDF* in order to obtain a good estimate and realistic limits for calibration of this model parameter as well as to predict changes during the melt season in case of forecasting or due to the effects of climate change.

**2. Test site and datasets**

**2.1 Test Site**

The test site locates in the Dreisäulerbach catchment, which is a part of the Isar River system and lies in the sub-alpine region of Bavaria in the Ammergauer Alps, Germany. Dreisäulerbach catchment approximately lies between latitudes 47°34'55"– 47°35'05" North and longitudes 10°56'40"–10°57'07"

East. It covers an area of about 2.3 km$^2$ and has a mean hypsometric elevation of just over 1200 m a.s.l. (Figure 1). The elevation ranges from about 950 m a.s.l. at Linderhof gauging station up to 1768 m a.s.l. at the Hennenkopf.

The area is mostly characterized by south facing slopes, but also contains northern slopes in southern parts of the catchment. The catchment is densely forested which during the winter season is fully snow-

covered. The mean annual temperature in the observation period (i.e. November 2016 – May 2021) is about 5.8 °C and the long-term mean annual precipitation at the Ettal-Linderhof station of the Water Science Service Bavaria is reported to be 1676 mm (Kopp et al., 2019).

In order to observe the seasonal snow dynamics, snow measurement instruments in addition to a standard meteorological station have been installed at the Brunnenkopfhütte test site at an elevation of 1602 m

a.s.l. (see Figure 2). The installed station has various sensors including temperature, pressure, wind, solar radiation (incoming, outgoing), snow depth, snow scale, snowpack analyser and pluviometer. Table 1 summarises the observed monthly meteorological data at Brunnenkopfhütte station. Figure 3 presents the observed snow water equivalent (SWE) at the test site.

**Table 1** Observed monthly average meteorological data – (Brunnenkopfhütte: November 2016 – May 2021)

| Variables[1] | Jan | Feb | Mar | Apr | May | Jun | Jul | Aug | Sep | Oct | Nov | Dec |
|---|---|---|---|---|---|---|---|---|---|---|---|---|
| $T_a$ (°C) | -2.48 | -0.41 | 0.52 | 4.14 | 6.76 | 12.40 | 13.62 | 14.22 | 9.69 | 7.72 | 3.06 | 0.08 |
| $P$ (mm) | 230.2 | 147.3 | 138.8 | 115.1 | 188.0 | 185.4 | 216.5 | 241.5 | 183.7 | 162.4 | 107.2 | 195.9 |
| $u$ (ms$^{-1}$) | 1.08 | 1.01 | 1.10 | 0.97 | 0.71 | 0.60 | 0.59 | 0.59 | 0.60 | 0.79 | 1.02 | 1.00 |
| $RH$ (%) | 74.2 | 69.3 | 73.4 | 72.2 | 82.1 | 78.2 | 76.7 | 78.1 | 82.8 | 71.7 | 70.5 | 69.4 |
| $A$ (-) | 0.80 | 0.74 | 0.69 | 0.51 | 0.42 | - | - | - | - | - | 0.45 | 0.72 |
| $K_T$ (-) | 0.51 | 0.52 | 0.53 | 0.53 | 0.40 | 0.43 | 0.43 | 0.45 | 0.48 | 0.55 | 0.50 | 0.49 |
| $SR_{in}$ (W m$^{-2}$) | 61 | 97 | 148 | 200 | 181 | 207 | 200 | 185 | 150 | 119 | 68 | 51 |

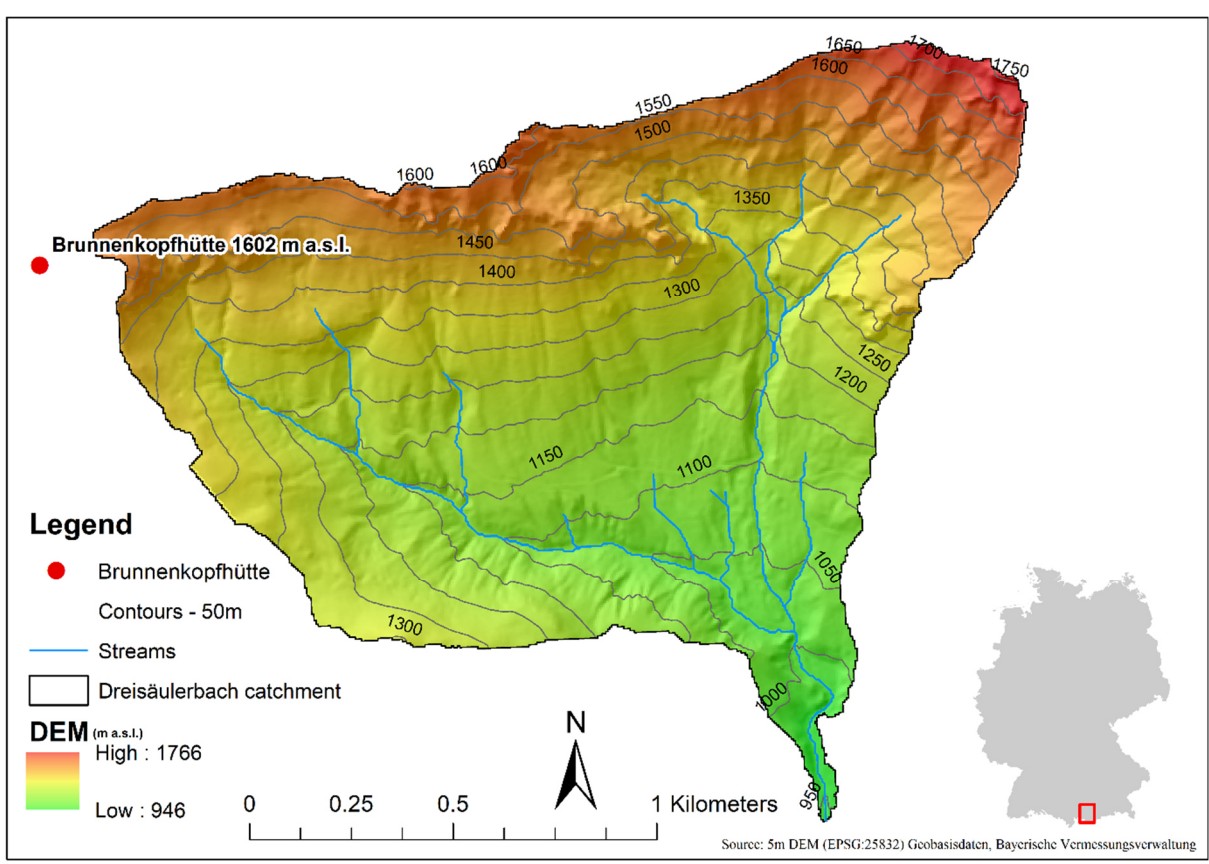

**Figure 1** Location of Brunnenkopfhütte automatic snow and weather station in the Dreisäulerbach catchment – German Alps

---

[1] $T_a$ = Air temperature

$P$ = Precipitation

$u$ = Wind speed

$RH$ = Relative Humidity

$A$= Albedo (*only considered when ground is snow covered*)

$K_T$ = Clearness index

$SR_{in}$ = Incoming shortwave radiation

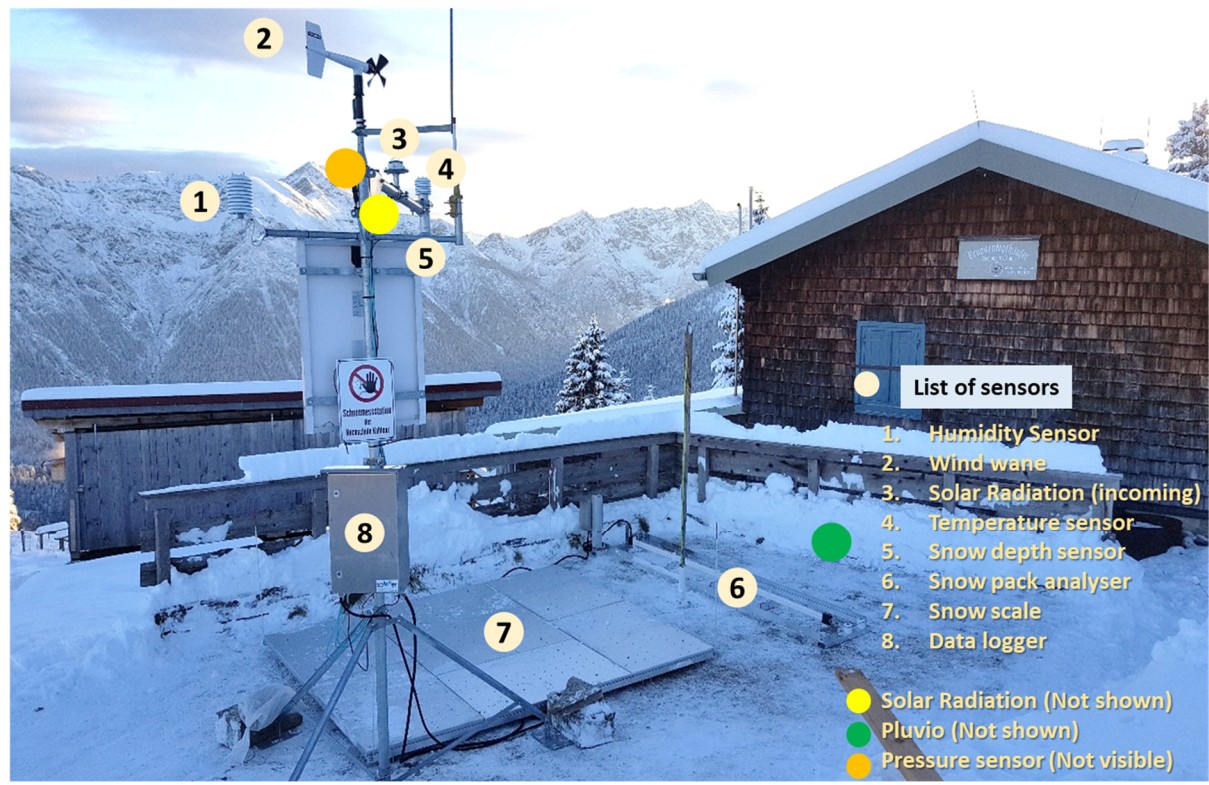

**Figure 2** Automatic snow and weather station at Brunnenkopfhütte in Ammergauer Alps [1602 m a.s.l.] (Image credit – Wolfgang Bogacki)

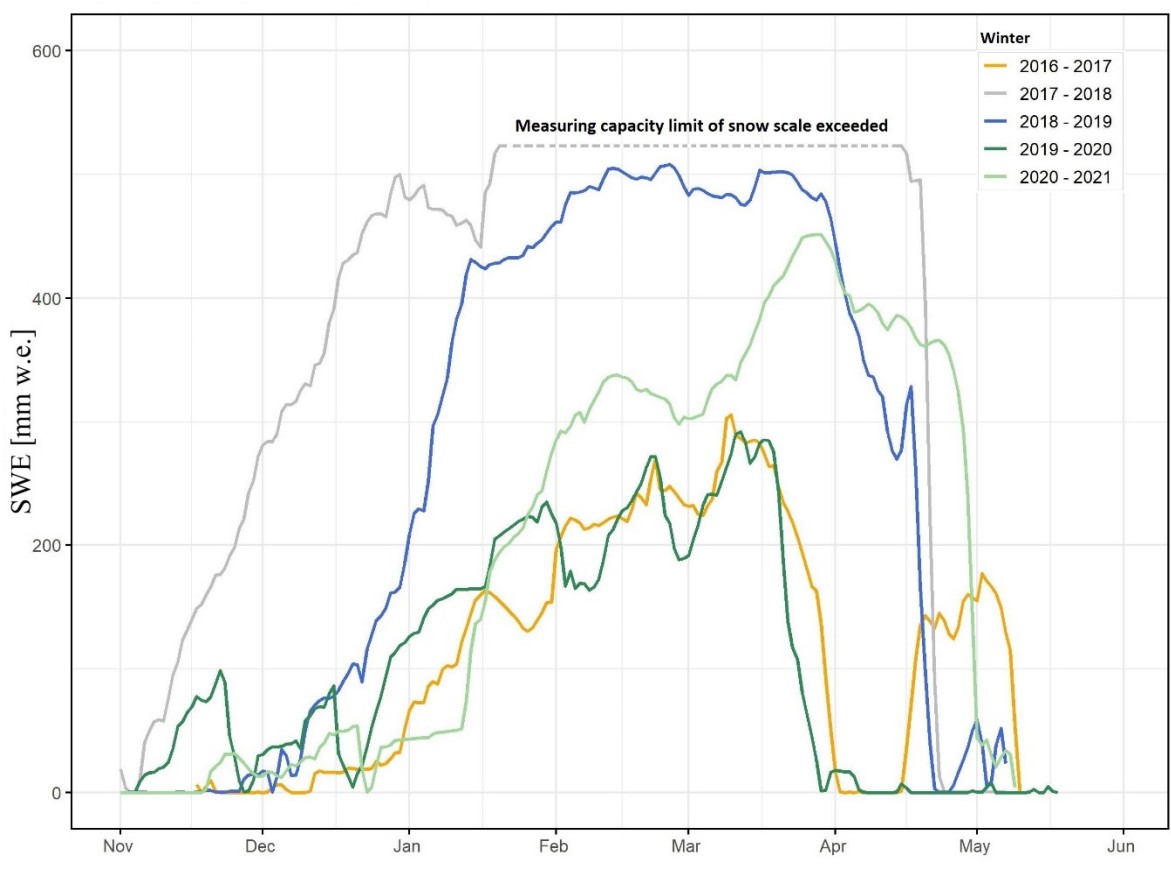

**Figure 3** Observed SWE [mm] at the Brunnenkopfhütte snow station (period: Winter 2016/2017 – 2020/2021)

**2.2 Datasets**

Present study utilises three different dataset. Data sources as well as aim of using these datasets are mentioned as follow:

1. We use observed hydro-meteorological datasets from a test site (i.e. Brunnenkopfhütte) with the aim to show how the *DDF* can be estimated for a specific site under naturally varying hydro-meteorological conditions.

2. In order to demonstrate the variation of the *DDF* over time, location, and altitude as well as its significance for temperature-index modelling, we use elevation zone-wise temperature data of
the Upper Jhelum Basin from a previous study (Bogacki and Ismail, 2016).

3. In the discussion section (Sec. 5), we perform a brief analysis in order to show the influence of climate change on the *DDF* in poorly monitored regions, for example Himalayas-Karakoram-Hindukush (i.e. Upper Indus Basin). In this specific analysis, projected changes in temperature are based on a previous study (Ismail et al., 2020). These projected changes in temperature are
the median of four GCMs (GFDL-ESM2M, HadGEM2-ES, IPSL-CM5A-LR, and MIROC5) that are driven by two representative concentration pathways (RCP2.6 and RCP8.5). This data is provided by the Inter-Sectoral Impact Model Intercomparison Project (ISIMIP) (Hempel et al., 2013; Frieler et al., 2017).

**3. Materials and methods**

The primary objective of this paper is to analyse the contribution of individual energy balance components to snowmelt, in order to better understand and probably to predict, how the lumped degree-day factor will vary with the season, latitude, altitude, and the actual meteorological conditions. In addition, we want to demonstrate following the approach of Walter et al., (2005), how these energy balance components can be estimated with minimal data requirements, as limited data availability is the
major reason to apply temperature-index respectively degree-day models.

**3.1    Degree-Day Factor**

The basic formulation of the degree-day method to calculate daily snowmelt depth $M$ (mm) multiplies the number of degree-days $T_{DD}$ (°C d) with the degree-day factor $DDF$ (mm °C$^{-1}$ d$^{-1}$) (Zingg, 1951; Braithwaite, 1995a; Rango and Martinec, 1995).

$$M = DDF \times T_{DD} \tag{1}$$

Degree-days $T_{DD}$ are only defined if a characteristic air temperature lies above a reference temperature $T_0$; otherwise, $T_{DD}$ is set to 0°C d. Typically, the freezing point $T_0 = 0$°C is chosen as reference temperature. Depending on the availability of temperature data, the characteristic air temperature is usually calculated as the mean of maximum and minimum daily air temperatures (Braithwaite, 1995a) or the mean of hourly observations (Rango and Martinec, 1995; DeWalle and Rango, 2008). But other

approaches like daily maximum temperature (Bagchi, 1983), integrating the positive part of a diurnal cycle (Ismail et al., 2015) or averaging the positive degree-day sum of $m$ daily observations (Braithwaite and Hughes, 2022) are also common.

By a simple re-arrangement of eq. (1) to

$$DDF = \frac{M}{T_{DD}} \tag{2}$$

the $DDF$ can be back-calculated for given degree-days $T_{DD}$, if the daily melt depth $M$ is known either by observation or by calculation. Likewise, the portion of the degree-day factor $DDF_i$ associated to the melt depth $M_i$ related to any of the individual energy balance components (see eq. (4)) can be determined.

The energy needed to melt ice at 0°C into liquid water at 0°C is defined by the latent heat of fusion of ice (333.55 kJ kg$^{-1}$). Thus the melt depth $M_i$ caused by an energy flux $Q_i$ (W m$^{-2}$) over a certain time-period $\Delta t$ (s) can be calculated from the relation (USACE, 1998; Hock, 2005)

$$M_i = \frac{Q_i}{\lambda\,\rho_w}\Delta t \cong 3.00 \times 10^{-6}\,Q_i\Delta t \tag{3}$$

where $\rho_w$ is the density of water at 0°C (999.84 kg m$^{-3}$). In the context of degree-day factor models, the time-period $\Delta t$ is usually taken as 1 day = 86400 s, though some authors (Hock, 1999; McGinn, 2012) have calculated degree-day factors also for sub-daily, e.g. hourly periods. According to the relation given in eq. (3), an energy flux of 1 W m$^{-2}$ for 1 day will result in a melt depth of 0.26 mm.

## 3.2 Energy Balance

In a unit area column of a snowpack, the energy flux available for snowmelt $Q_M$ can be calculated from the balance of energy fluxes entering or leaving the snowpack and the change in the internal energy stored in that column $\Delta Q$ (e.g. USACE, 1998)

$$Q_M = Q_S + Q_L + Q_H + Q_E + Q_G + Q_P - \Delta Q \tag{4}$$

where $Q_S$ and $Q_L$ are the net short- and longwave radiation, $Q_H$ is the sensible heat, $Q_E$ the latent energy of condensation or vaporization, $Q_G$ the heat conduction from the ground, and $Q_P$ the energy contained 205 in precipitation (all terms in W m$^{-2}$).

In the following sections, the individual components of the energy balance are discussed in more detail.

### 3.2.1 Shortwave Radiation

Shortwave radiation emitted from the sun is usually the largest source of energy input to the snowpack. The net energy flux $Q_S$ (W m$^{-2}$) entering the snowpack by absorption of shortwave radiation is

$$Q_s = (1 - A)S_i \tag{5}$$

where $A$ is the snow albedo (–) and $S_i$ the incident solar radiation (W m$^{-2}$) on the snow surface. A widely used approach to determine the incident solar radiation on earth's surface is the introduction of a clearness index $K_T$ (–)

$$S_i = K_T S_0 \tag{6}$$

where $S_0$ is the mean daily potential extra-terrestrial solar radiation (W m$^{-2}$) that would insolate a horizontal surface on the earth's ground if no atmosphere would be present.

**Potential insolation at the top of atmosphere**

The potential insolation, which is only dependent on the changing position of the sun during the year in relation to the geographic location of the incident point on the earth's surface, can be calculated from the equation (Masters, 2004)

$$S_0 = G_s \frac{1}{d_r^2} \frac{1}{\pi} (\cos(\emptyset) \cos(\delta) \cos(\omega_s) + \omega_s \sin(\emptyset) \sin(\delta)) \tag{7}$$

where $G_S$ is the solar constant (W m$^{-2}$), $d_r$ the relative distance earth to sun (–), $\phi$ the geographic latitude (rad) of the incident point, $\delta$ the solar declination (rad), and $\omega_s$ the sunrise hour angle (rad). The solar constant $G_S$ is slightly varying with the occurrence of so-called sunspots. Measurements by Kopp and Lean, (2011) indicate a present value of about 1361 W m$^{-2}$.

Both sun position variables, the relative distance earth to sun and the solar declination, can be calculated quite exactly by rigorous astronomical algorithms (Meeus, 1991; Reda and Andreas, 2004) but for non-astronomical purposes, more simple formulas are sufficiently accurate. The relative distance earth to sun, which is varying over the year due to the elliptical orbit of the earth, can be approximated by (Masters, 2004)

$$\frac{1}{d_r^2} \approx 1 + 0.034 \cos\left(\frac{2\pi . J}{365}\right) \tag{8}$$

where $J$ is the day number, with $J = 1$ on January 1$^{st}$. The solar declination can be obtained from the sinusoidal relationship

$$\delta \approx 0.409 \sin\left(\frac{2\pi}{365}(J - 81)\right) \tag{9}$$

that puts the spring equinox on day $J = 81$. Knowing the solar declination $\delta$, the sunrise hour angle $\omega_s$ can be calculated from

$$\cos \omega_s = -\tan(\emptyset) \tan(\delta) \tag{10}$$

On the northern hemisphere the maximum extra-terrestrial radiation occurs at the summer solstice with a fairly identical mean daily energy flux of about 480 W m$^{-2}$ over latitudes 30° – 60° North, as the sun's lower altitude angle at higher latitudes is compensated by longer daylight hours. On the contrary, minimum extra-terrestrial radiation at the winter solstice varies strongly with latitude, e.g. 227 W m$^{-2}$ at 30° and only 24 W m$^{-2}$ at 60° North.

**Clearness Index**

When the solar radiation passes through the atmosphere, it is partly scattered and absorbed. While even on a clear day only about 75% of the incoming radiation reaches the ground, by far the largest reflection is caused by clouds. A vast number of solar radiation models exist that parameterise this effect, which is denoted as clearness index $K_T$ or atmospheric transmissivity $\tau$, as a function of meteorological variables. For a review see e.g. Evrendilek and Ertekin (2008), Ahmad and Tiwari (2011), or Ekici (2019).

A fundamental and widely used solar radiation model which is proposed in the context of evapotranspiration calculations (Allen et al., 1998) is the Ångström-Prescott model, that relates the clearness index to the relative sunshine duration

$$K_T = \frac{S_i}{S_0} = a + b\frac{n}{N} \tag{11}$$

with $n$ is the actual and $N$ the maximal possible duration of sunshine (hr) where the latter can be calculated from the sunrise hour angle $\omega_s$ by

$$N = \frac{24}{\pi}\omega_s \tag{12}$$

The parameters $a$ and $b$ in eq. (11) are regression parameters, that usually have to be fitted to observed global radiation. In case no actual solar radiation data is available, the values $a = 0.25$ and $b = 0.50$ are recommended (Allen et al., 1998). Though the Ångström-Prescott model has the disadvantage that the parameters have to be fitted and the actual duration of sunshine has to be observed, it has the benefit that both parameters allow for a direct physical interpretation. The parameter $a$ represents the clearness index $K_T$ on overcast days ($n = 0$), while their sum $a + b$ gives the clearness index on clear days ($n = N$).

In common situation in remote mountainous regions, when only temperature data is available, another group of solar radiation models can be utilised, which uses the difference between daily maximum and minimum air temperature $\Delta T$ (°C) as a proxy for cloud cover, because clear sky conditions result in a higher temperature amplitude between day and night than under overcast conditions. Typical models are the exponential approach proposed by Bristow and Campbell (1984) and its later modifications or the simple empirical equation by Hargreaves and Samani (1982)

$$K_T = k_H \sqrt{\Delta T} \tag{13}$$

with the empirical coefficient $k_H = 0.16$ for inland and $k_H = 0.19$ for coastal locations. Since the influence of cloud cover on the clearness index and thus on the *DDF* can be illustrated much more directly by Ångström-Prescott type models, this model type is further on used in the paper.

It is obvious, that the attenuation of extra-terrestrial solar radiation is a function of the distance the rays have to travel through the atmosphere, as absorption and scattering occurs all along the way. Several solar radiation models consider altitude as a variable, of which the models below were calibrated including high altitude stations and are of Ångström-Prescott type, thus the altitude effects can be compared directly.

Jin et al. (2005):

$$\text{(a)} \qquad K_T = (0.0855 + 0.0020\emptyset + 0.030z) + 0.5654\frac{n}{N} \tag{14}$$

$$\text{(b)} \qquad K_T = (0.1094 + 0.0014\emptyset + 0.0212z) + (0.5176 + 0.0012\emptyset + 0.0150z)\frac{n}{N} \tag{15}$$

Rensheng et al. (2006):

$$K_T = (0.122 + 0.001\emptyset + 0.0257z) + 0.543\frac{n}{N} \tag{16}$$

Liu et al. (2019):

$$K_T = (0.1755 + 0.0136z) + (0.5414 + 0.0117z)\frac{n}{N} \tag{17}$$

For all models, $z$ is the altitude (km) and $\phi$ the latitude (deg).

In order to evaluate the altitude effect separately from other parameters, the clearness index $K_T$ splits into two components

$$K_T = K_{T_0} \cdot K_z \tag{18}$$

where $K_{T_0}$ is the clearness index at $z = 0$ m a.s.l. and $K_z$ is a clearness altitude factor (–) which represents the increase of $K_T$ with altitude relative to $K_{T_0}$. At sea level, $K_Z = 1$ for all models and all values of relative sunshine duration $n/N$. Though the clearness altitude factors $K_Z$ obtained from eq. (14) – (17) are different for each equation, they all show a linear increase with altitude, the slope of which depends on the cloudiness (see Figure 4).

Using, for example eq. (15), at sea level the clearness factor $K_T = K_{T_0}$ would be 0.15 and 0.72 for overcast and clear sky conditions respectively, while $K_T$ increases to 0.19 and 0.79 at an altitude of $z =$

2000 m a.s.l. The resulting clearness altitude factors $K_z$ are 1.27 and 1.10 respectively. It should be noted, that although $K_z$ is higher for overcast than for clear sky conditions, the absolute increase of the clearness index $K_T$ with altitude is higher under clear sky conditions.

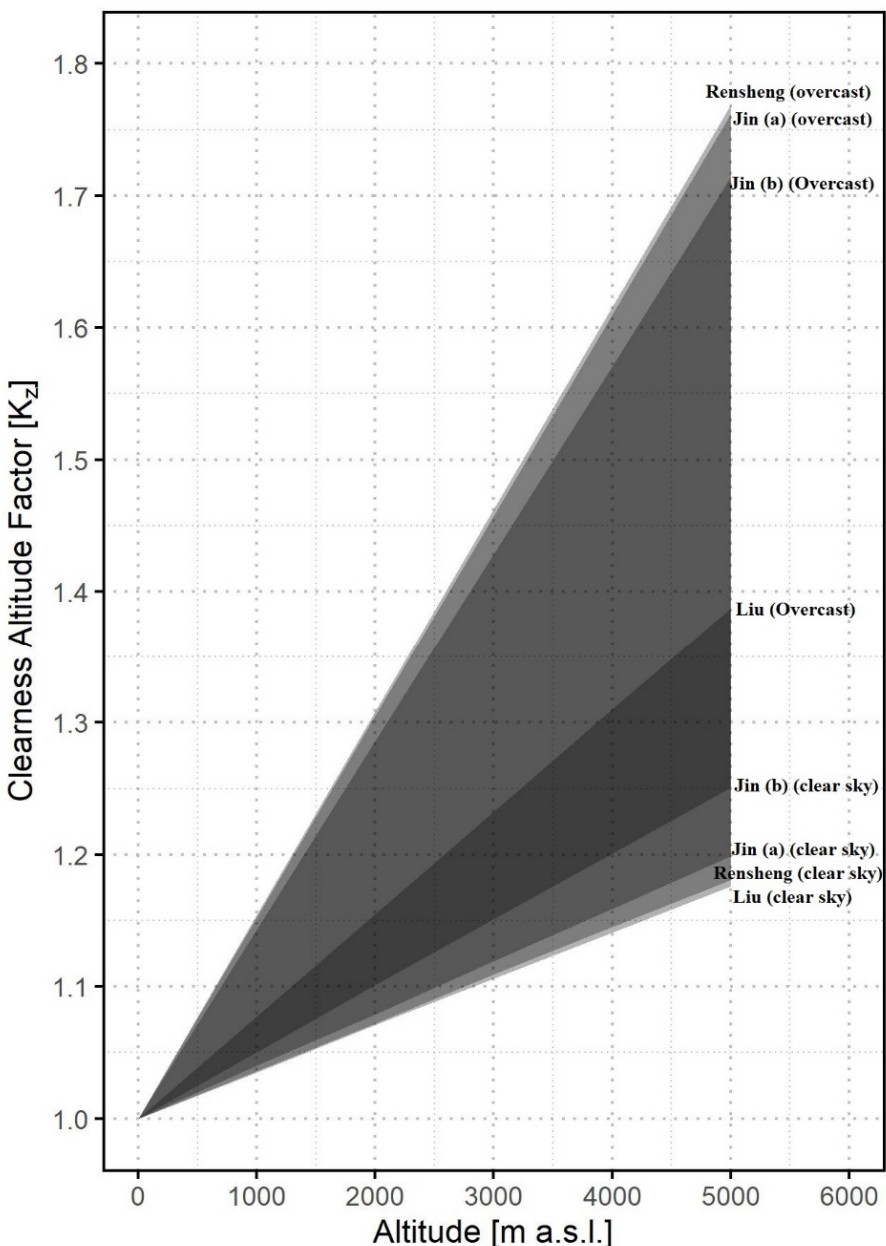

Figure 4 Clearness altitude factor $K_z$ for different altitudes ranges, based on different models presented in equations (14 – 17, i.e. Jin (a), Jin (b), Rensheng, and Liu) for latitude 45° where applicable

**Albedo**

While the albedo of fresh snow is well above 0.9 (Hock, 2005), indicating that most of the shortwave radiation is reflected, it may drop significantly within a few days due to snow metamorphism. Well aged snow generally has an albedo in the range of 0.4 – 0.5 (Anderson, 2006). Snow albedo is primarily dependent on the grain size of the snow crystals near the surface but also on aerosols in the snow and dust deposits. Respective snow albedo models are proposed e.g. by Wiscombe and Warren (1980) and

Warren and Wiscombe (1980). However, because of their data requirements, rather surrogate exponential decay models as formulated by USACE, (1956) are commonly in use, which assume the decrease of albedo as a function of time after the last significant snowfall. For example Walter et al., (2005) use the empirical relationship

$$A_n = 0.35 - (0.35 - A_{max}).\,exp\left[-\left(0.177 + \ln\left(\frac{A_{max} - 0.35}{A_{n-1} - 0.35}\right)^{2.16}\right)\right]^{0.46} \qquad (19)$$

where $A_{n-1}$ is the albedo of the previous day and $A_{max}$ is the maximum albedo (~0.95) of fresh snow. Following eq. (19), the snow albedo will decrease from 0.95 to 0.52 after 10 days and to 0.43 after 30 days if no new snowfall occurs.

### 3.2.2  Longwave Radiation

The net longwave radiation flux over the snow surface $Q_L$ (W m$^{-2}$) is the balance between incoming longwave radiation that is emitted by the atmosphere $Q_{L,in}$ (W m$^{-2}$) and outgoing radiation from the snowpack $Q_{L,out}$ (W m$^{-2}$).

$$Q_L = Q_{L,in} - Q_{L,out} \qquad (20)$$

Longwave radiation is a function of the temperature of the emitting body and can be calculated with the Stefan-Boltzmann law

$$L = \varepsilon\,\sigma\,T^4 \qquad (21)$$

where $L$ is the radiative flux (W m$^{-2}$), $\varepsilon$ and $T$ are the emissivity (–) and the absolute temperature (K) of the emitting body, and $\sigma$ is the Stefan-Boltzmann constant (5.67×10$^{-8}$ W m$^{-2}$ K$^{-4}$).

In particular fresh snow is nearly a perfect blackbody with respect to longwave radiation, thus it has a high emissivity of 0.99 (Warren, 1982; USACE, 1998; Anderson, 2006). For old snow, Brutsaert (1982) gives an emissivity value of 0.97. Given a melting snowpack having a surface temperature of 0°C, the outgoing energy flux can be taken as constant with $Q_{L,out}$ ~310 W m$^{-2}$.

For the atmospheric longwave radiation, usually the air temperature $T_a$ (K) is used in eq. (21). However, while the snowpack longwave emissivity is virtually a constant, the emissivity of the atmosphere is highly variable. Typical values under clear sky conditions range from 0.6 – 0.8, primarily depending on air temperature and humidity (Anderson, 2006) whereas for overcast conditions it can be close to 1.0.

A number of empirical and more physically based approaches exist to estimate atmospheric longwave emissivity from standard meteorological data (see Hock, 2005 for a discussion). For clear sky conditions, Brutsaert (1975) developed a theoretically based formula depending on air temperature and vapour pressure measured at screen level

$$\varepsilon_{ac} = 1.24 \left(\frac{p_v}{T_a}\right)^{\frac{1}{7}} \tag{22}$$

where $\varepsilon_{ac}$ is the clear sky longwave emissivity (–), $p_v$ the actual vapour pressure (hPa), and $T_a$ the air temperature (K). Later, Brutsaert reconciled eq. (22) with an empirical approach proposed by Swinbank (cited in Brutsaert, 1982)

$$\varepsilon_{ac} = 9.2 \times 10^{-6} \, T_a^2 \tag{23}$$

that considers the strong correlation between vapour pressure and air temperature, thus only air temperature is needed as input variable. Using above relation, at an air temperature of 10 °C the atmospheric longwave radiation flux into the snowpack amounts to $Q_{L,in} = 281$ W m$^{-2}$ under clear sky conditions, which is less than the outgoing flux of 310 W m$^{-2}$, i.e. the snowpack will lose energy in this situation.

The variability of atmospheric emissivity due to cloud cover, which increases the longwave emissivity, is significantly higher than variations under clear sky conditions. Monteith and Unsworth (2013) give the simple linear relationship.

$$\varepsilon_a = (1 - 0.84c)\varepsilon_{ac} + 0.84c \tag{24}$$

where $\varepsilon_a$ is the atmospheric longwave emissivity, $c$ the fraction of cloud cover (–), and $\varepsilon_{ac}$ is calculated by eq. (22) or eq. (23). For overcast conditions and an air temperature of 10 °C, eq. (24) yields an atmospheric emissivity of 0.96, which results in an atmospheric longwave radiation flux of $Q_{L,in} = 351$ W m$^{-2}$ and thus a positive flux of $Q_L = 41$ W m$^{-2}$ into the snowpack.

Although cloud cover is difficult to parameterise, as clouds can be highly variable in space and time and their effects on radiation depend on the different cloud genera, a strong correlation between cloud cover and sunshine duration is obvious. Doorenbos and Pruitt, (1977) give a tabulated relation between cloudiness $c$ and relative sunshine hours $n/N$ (see eq. (11)), that can be fitted by the quadratic regression

$$c = 1 - 0.5544 \, \frac{n}{N} - 0.5483 \left(\frac{n}{N}\right)^2 \tag{25}$$

Nevertheless, in simple sky models usually a linear relation between cloudiness and relative sunshine hours is applied as a first approximation (e.g. Brutsaert, 1982; Annandale et al., 2002; Pelkowski, 2009) which, as Badescu and Paulescu, (2011) showed by using probability distributions to develop relations between cloudiness and relative sunshine hours, is a first good estimate.

### 3.2.3 Sensible Heat Exchange

Sensible heat exchange describes the energy flux due to temperature differences between the air and the snow surface while air is permanently exchanged by wind turbulences. A frequent approach to

parameterise turbulent heat transfer is the aerodynamic method, that explicitly includes wind speed as a variable (Braithwaite et al., 1998; Lehning et al., 2002; Hock, 2005)

$$Q_H = \rho_a c_p C_H u (T_a - T_s) \qquad (26)$$

where $\rho_a$ is the air density (kg m$^{-3}$), $c_p$ the specific (isobaric) heat capacity of air (1006 J kg$^{-1}$ °C$^{-1}$), $C_H$ the exchange coefficient for sensible heat (–), $u$ the mean wind speed (m s$^{-1}$), $T_a$ the air temperature (°C), and $T_s$ the temperature at the snow surface (°C).

The density of air $\rho_a$ is a function of atmospheric pressure, air temperature, and humidity

$$\rho_a = \frac{M_d [p - (1-e) p_v]}{R T_a} \qquad (27)$$

where $p$ is the atmospheric pressure (Pa), $p_v$ the vapour pressure (Pa) (see eq. (32)), $T_a$ the air temperature (K), $M_d$ the molar mass of dry air (0.02897 kg mol$^{-1}$), $R$ the universal gas constant (8.31446 J mol$^{-1}$ K$^{-1}$) and $e$ the ratio of molar weights of water and dry air = 0.622. At usual air temperatures humidity has only a minor effect on the air density.

The decrease of atmospheric pressure with altitude $z$ (m a.s.l.) can be estimated by the isothermal barometric formula

$$p(z) = p_0 \, exp\left(-\frac{g M_d}{R T_a} z\right) \qquad (28)$$

where $p_0$ is the atmospheric pressure at sea level (Pa) and $g$ the gravitational acceleration (m s$^{-2}$). At an air temperature of 0 °C and a standard atmospheric pressure at sea level of 101.325 kPa, the air density is 1.29 kg m$^{-3}$ while e.g. at an altitude of 2000 m a.s.l. the atmospheric pressure reduces to 78.9 kPa and the air density becomes 1.01 kg m$^{-3}$.

The exchange coefficient $C_H$ can be approximated with (Campbell and Norman, 1998)

$$C_H = \frac{k^2}{ln\left(\frac{z_u}{z_m}\right) ln\left(\frac{z_T}{z_h}\right)} \qquad (29)$$

where $k$ is the von Kármán's constant 0.41 (–), $z_u$ and $z_T$ the height of wind and temperature observation above the snow surface (m), $z_m$ the momentum roughness parameter, and $z_h$ the heat roughness parameter. For a snow surface, the roughness parameters are given by Walter et al., (2005) as $z_m$ ~ 0.001 m and $z_h$ ~ 0.0002 m.

Eq. (29) is equivalent to the calculation of aerodynamic resistance in the Penman-Monteith equation (Allen et al., 1998) when applying a zero plane displacement for the snow surface and assumes neutral stability conditions, i.e. temperature, atmospheric pressure, and wind velocity distributions follow nearly

adiabatic conditions. Otherwise, diabatic correction factors (see Campbell and Norman, 1998) have to be applied.

As can be seen from eq. (26), the sensible heat component depends mainly on wind speed and temperature. During stable clear weather periods with typically light winds, the turbulent exchange is smaller on average than the radiation components. For example, a wind speed of 1 m s$^{-1}$ and an air temperature of 5 °C will result in a sensible heat flux of about 15.5 W m$^{-2}$. However, at warm rain events or at Föhn conditions with strong warm winds, turbulent exchange can significantly contribute to the melt process. For example a Föhn event of 14 hours duration on 8$^{th}$ December 2006 at Altdorf (Switzerland, 440 m a.s.l.) with an average air temperature of about 16 °C, average relative humidity of 37% and average wind speed of 14.6 m s$^{-1}$ resulted in a mean sensible heat flux of about 700 W m$^{-2}$ during that duration.

### 3.2.4 Latent Energy of Condensation or Vapourisation

The latent energy exchange reflects the phase change of water vapour at the snow surface, either by condensation of vapour contained in the air or by vapourisation of snow. Thus, it can either warm or cool the snowpack (Harpold and Brooks, 2018). The energy flux is dependent on the vapour gradient between the air and the snow surface and is, like the sensible heat exchange, a turbulent process that increases with the wind speed. Thus, the aerodynamic formulation is analogously to eq. (26)

$$Q_E = \rho_a \lambda_v C_E u (q_a - q_s) \tag{30}$$

where $\lambda_v$ is the latent heat of vapourisation of water at 0°C (2.501×10$^6$ J kg$^{-1}$), $C_E$ the exchange coefficient for latent heat (–) which is assumed to be equal to the exchange coefficient for sensible heat $C_H$, $q_a$ the specific humidity of the air (–), and $q_s$ the specific humidity at the snow surface (–).

The specific humidity $q_a$ can be derived from measurements of relative humidity or dew point temperature. In cases where such data is not available, Walter et al., (2005) approximate the dew point temperature by the minimum daily temperature. For any air temperature $T$ (°C), the saturation vapour pressure $p_s$ (Pa) can be calculated by an empirical expression known as the Magnus-Tetens equation in the general form (Lawrence, 2005)

$$p_s = C \, e^{\frac{A \, T}{B+T}} \tag{31}$$

where $A$, $B$, and $C$ are coefficients e.g. after Allen et al., (1998) $A = 17.2694$, $B = 237.3$ °C, $C = 610.78$ Pa. At the snow surface, according to Lehning et al., (2002) the air temperature can be assumed equal to the snow surface temperature and eq. (31) is applied with coefficients for saturation vapour pressure over ice $A = 21.8746$, $B = 265.5$ °C, $C = 610.78$ Pa (Murray, 1967). At a temperature of 0°C, both coefficient sets yield the same saturation vapour pressure of $p_s = 611$ Pa.

Knowing the relative humidity $\psi$ (-) and the saturation vapour pressure $p_s$ at a given air temperature, the actual vapour pressure $p_v$ (Pa) can be calculated through the relation

$$p_v = \psi \, p_s \tag{32}$$

and subsequently the respective specific humidity by

$$q = \frac{e \; p_v}{p - (1-e)p_v} \approx \frac{e}{p} p_v \tag{33}$$

with $p$ the atmospheric pressure (Pa) and $e$ the ratio of molar weights of water and dry air $= 0.622$ as in eq. (27). Assuming melting conditions with a snow temperature $T_s = 0\ °C$ and saturated vapour conditions, the vapour pressure at the snow surface is $p_{v,snow} = p_s(0\ °C) = 611$ Pa. While at positive air temperatures the sensible heat flux is always warming the snowpack, the latent heat flux can cool the snow by vapourisation if the relative humidity of the air is low. Even when assuming a relative humidity of 100% the latent heat flux into the snowpack will be comparatively small if wind speed is low, e.g. about 13 W m$^{-2}$ at an air temperature of 5 °C and a wind speed of 1 m s$^{-1}$.

### 3.2.5 Ground Heat

Heat conduction from the ground into the snowpack is small and can be in general neglected except when first snow falls on warm ground (Anderson, 2006). If the snowpack is well established, due to the low thermal conductivity of snow the heat flux across the soil-snow interface becomes independent of air temperature fluctuations and depends only on the thermal conductivity of the soil and the temperature gradient in the upper soil layer. USACE, (1998) gives a range between $0 – 5$ W m$^{-2}$ for constant daily values. DeWalle and Rango, (2008) approximate a flux of 4 W m$^{-2}$ assuming a soil temperature of 1 °C at a depth of 0.5 m, and a soil thermal conductivity of 2 W m$^{-1}$ °C$^{-1}$ that is at the higher end of the range of $0.2 – 2$ W m$^{-1}$ °C$^{-1}$ given by Oke, (1987). It has to be noted, that the soil temperature gradually approaches the snowpack temperature during the winter (USACE, 1956; Marks et al., 1992), thus ground heat conduction will generally decrease. Own measurements of soil temperature show a similar behaviour. Soil temperature dropped from $1 – 3$ °C shortly after establishment of the snowpack to $> 0 – 1$ °C after about a month and then stayed constant until final melt. Since the contribution of ground heat to the *DDF* is negligible, it is not considered in the further analysis.

### 3.2.6 Precipitation Heat

The heat transfer into the snowpack by lowering rain's temperature, that is usually assumed to be equal to the air temperature $T_a$ (°C), down to the freezing point at 0°C can be estimated as

$$Q_P = c_w \, P \, T_a \tag{34}$$

where $c_w$ is the specific heat capacity of water (4.2 kJ kg$^{-1}$ °C$^{-1}$) and $P$ is the daily rainfall depth (kg m$^{-2}$ d$^{-1}$). The energy input from precipitation is usually quite small and even during extreme weather

conditions, like heavy warm rain storms with temperatures of 15°C and a precipitation depth of 50 mm, that may occur e.g. during early winter in the alps, the mean daily energy flux from rain would be a moderate 36.5 W m$^{-2}$. In addition, such events are rare and of limited duration.

### 3.2.7 Change in Internal Energy

The rate of change in the energy stored in the snowpack $\Delta Q$ (W m$^{-2}$) represents the internal energy gains and losses due to changes in the snowpack's temperature profile and due to phase changes, i.e. melting of the ice portion or refreezing of liquid water in the snowpack. Until the snowpack temperature is isothermal at 0 °C, any melt produced in the surface layer that exceeds the liquid water holding capacity of the porous snow matrix will percolate downward and will be captured and refrozen in colder lower layers. This internal mass and energy transport process absorbs at least parts of the incoming energy, which reduces the energy available for melt and thus will reduce the actual *DDF*.

Under data scarce conditions and particularly when only daily data is available, it is difficult to properly quantify the change in the internal energy of the snowpack (see discussion in Sec. 5.2.1). Therefore, the present paper focusses on melt periods when the snowpack is 'ripe', i.e. the temperature is isothermal at 0 °C and the residual volumetric water content of about 8% (Lehning et al., 2002) is filled with liquid water. This assumption is not a limitation when analysing the contribution of each individual energy flux component towards a resulting *DDF* as presented in following sections, but the additional energy needed for 'warming' the snowpack has to be taken into account when estimating the total *DDF* if a snowpack is not 'ripe' (see Figure 11).

### 4. Results

In this section, the contribution of each energy flux component $Q_i$ to the lumped daily *DDF* is presented. For this purpose, the respective melt depth $M_i$ is calculated according to eq. (3) and further converted into the corresponding degree-day factor component DDF$_i$ using eq. (2). For the following exemplary calculations, the air temperature is assumed to stay always above 0 °C, thus degree-days $T_{DD}$ (°C d) in eq. (2) have the same numerical value as the daily average air temperature $T_a$ (°C) used in the calculation of several energy flux components.

Besides demonstrating the dependency of the *DDF* components on decisive parameters of the energy flux components, the presented tables and graphs, which are based on the relationships given in section 3, can be used to estimate the *DDF* component values in case either observed data is not available or not sufficient for more sophisticated approaches. It should be noted that parameters are normalised where applicable, i.e. set to hypothetical values like clearness index $K_T = 1$ or wind speed $u = 1.0$ m s$^{-1}$, thus final *DDF* values can be obtained by multiplying the given figures by the actual values of those parameters. Furthermore, all results are based on the assumption that the snowpack is isothermal at 0°C and in fully ripe state.

### 4.1 Shortwave radiation component - DDF$_S$

Shortwave radiation induced melt is usually considered the largest *DDF* component especially at higher elevations as well as under dry climates. The net energy flux $Q_S$ is calculated using eq. (5), which consists of three factors (a) latitude, (b) albedo, and (c) clearness index $K_T$. The dependency of DDF$_S$ on these factors is demonstrated in Figure 5 for the period between winter solstice (21st December) and summer solstice (21st June). As shortwave radiation is independent of air temperature and hence of

degree-days, the corresponding melt is divided by a hypothetical degree-day value of 1 °C d to arrive at DDF$_S$ values as presented. In case of actually higher degree-days, the given DDF$_S$ values have to be divided accordingly.

Figure 5 (a) shows the variation of DDF$_S$ depending on latitude for the range 30° – 60° North, while albedo ($A = 0$) and clearness index ($K_T = 1$) are set constant. Obviously, there is a significant difference

in DDF$_S$ for different latitudes around the winter solstice due to solar inclination, making latitude the predominant factor for DDF$_S$ at this time of the year. However, around the summer solstice, DDF$_S$ has nearly the same value at different latitudes because the lower solar angle at higher latitudes is counterweighted by a larger hour angle, i.e. longer sunlight hours. Thus, with the progress of the melting season the factors albedo and clearness index become more important than latitude.

Figure 5 (b) shows the influence of albedo on the DDF$_S$ at a given latitude (Brunnenkopfhütte test site – latitude 48°) and normalised constant clearness index ($K_T = 1$). Snow albedo is varied between 0.9 – 0.4 covering the range between fresh and well-aged snow. As to be expected, the influence of albedo increases with increasing incoming solar radiation towards the summer solstice. A good estimate of albedo is therefore much more important when the snowmelt season progresses than in early spring. If

for example the same degree-day value of 10 °C d is assumed on 21st March and on 21st May, the difference in DDF$_S$ between fresh ($A = 0.9$) and aged ($A = 0.4$) snow would be 0.8 and 4.6 mm °C$^{-1}$ d$^{-1}$ in March compared to 1.2 and 7.1 mm °C$^{-1}$ d$^{-1}$ in May respectively.

The dependency of DDF$_S$ on the clearness index $K_T$ is shown in Figure 5 (c). As also evident from eq. (6), DDF$_S$ under clear sky ($K_T = 0.75$) is always higher than under overcast conditions ($K_T = 0.25$).

Similar to albedo, the influence of the clearness index becomes more pronounced, and thus the assessment of clearness conditions more important, with increasing solar angle when the snowmelt season progresses.

The influence of altitude on DDF$_S$ in terms of increasing $K_T$ values can be assessed by multiplying a clearness index $K_{T_0}$ at sea level, which may be obtained by any of the numerous solar radiation models,

with a clearness altitude factor $K_z$ (see eq. (18)). Figure 4 shows the range of clearness altitude factors for latitude 45° derived from eq. (14) – (17) . All $K_z$ values show a linear increase with altitude, with the slope depending on cloudiness. It should be noted that although the increase of $K_z$ relative to $K_{T_0}$ is higher under overcast than under clear sky conditions, the absolute increase of the clearness index $K_T$ with altitude is larger for clear sky conditions (see Sec. 3.2.1). When using the intersection of all models

and sky conditions, which is indicated by the dark grey area in Figure 4, in order to get one overall rough estimate of $K_z$ for all conditions, the clearness altitude factor and thus the resulting DDF$_S$ is found to increase by about 6.4% per each 1000 m of altitude.

(a)  (b)

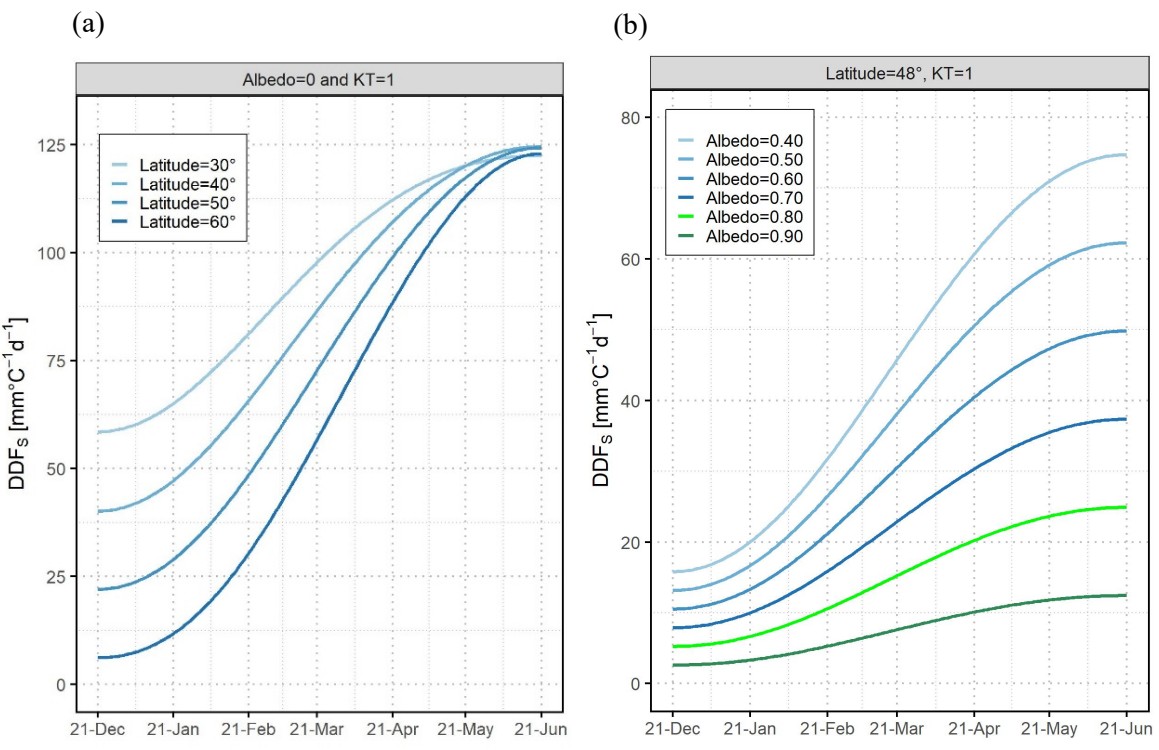

(c)

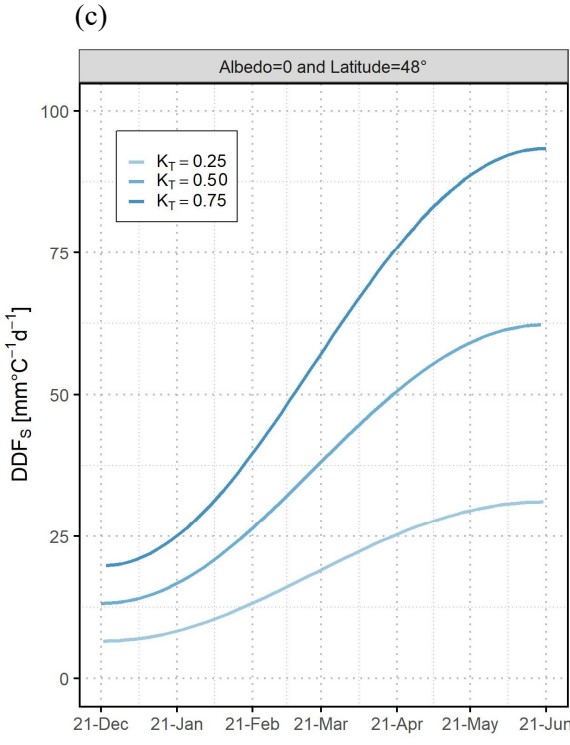

Figure 5 Variation of solar radiation based DDF$_S$ for a degree-day value of 1°C d for (a) different latitudes under constant snow albedo and clearness index; (b) snow albedos under constant

latitude and clearness index; (c) different clearness indices under constant latitude and snow albedo – The latitude = 48° corresponds to the location of Brunnenkopfhütte test site.

**4.2 Longwave radiation component - DDF$_L$**

The net longwave energy flux $Q_L$ is calculated using eq. (21), in which the outgoing radiation from the snowpack can be assumed as constant. Thus, the contribution of longwave radiation component DDF$_L$ is mainly dependent on air temperature and the emissivity of the atmosphere, in particular cloudiness conditions. Figure 6 and Table 2 present the DDF$_L$ as a function of degree-days $T_{DD}$ and cloudiness. For a wide range of degree-days especially in conjunction with low cloudiness, the outgoing longwave energy flux is higher than the incoming, resulting in a theoretically negative degree-day factor that will reduce the total $DDF$. This means that the DDF$_L$ component under clear sky conditions usually is rather contributing to a cooling of the snowpack than to melting. Under overcast conditions, the DDF$_L$ is relatively constant around 1 mm °C$^{-1}$ d$^{-1}$ with a maximum value of 1.3 mm °C$^{-1}$ d$^{-1}$ at $T_{DD}$ = 20 °C d. Although this contribution to the total $DDF$ is small compared to the shortwave radiation component DDF$_S$, it can be of importance at the onset of snowmelt in early spring, when the solar radiation is still low and the albedo of fresh snow is high.

**Table 2**     Longwave radiation component (DDF$_L$) [mm °C$^{-1}$ d$^{-1}$] for selected cloudiness [%] and degree-days [°C d]

| | DDF$_L$ | | | | | | | | | | |
| --- | --- | --- | --- | --- | --- | --- | --- | --- | --- | --- | --- |
| | Cloudiness | | | | | | | | | | |
| $T_{DD}$ [°C d] | 0% | 10% | 20% | 30% | 40% | 50% | 60% | 70% | 80% | 90% | 100% |
| 1 | -19.08 | -17.17 | -15.26 | -13.35 | -11.45 | -9.54 | -7.63 | -5.72 | -3.81 | -1.90 | 0.01 |
| 2 | -8.89 | -7.94 | -6.99 | -6.04 | -5.09 | -4.14 | -3.19 | -2.24 | -1.29 | -0.33 | 0.62 |
| 3 | -5.49 | -4.86 | -4.23 | -3.59 | -2.96 | -2.33 | -1.70 | -1.07 | -0.44 | 0.19 | 0.82 |
| 4 | -3.78 | -3.31 | -2.84 | -2.37 | -1.90 | -1.42 | -0.95 | -0.48 | -0.01 | 0.46 | 0.93 |
| 5 | -2.75 | -2.38 | -2.00 | -1.63 | -1.25 | -0.88 | -0.50 | -0.13 | 0.25 | 0.62 | 1.00 |
| 6 | -2.06 | -1.75 | -1.44 | -1.13 | -0.82 | -0.51 | -0.20 | 0.11 | 0.43 | 0.74 | 1.05 |
| 7 | -1.57 | -1.30 | -1.04 | -0.77 | -0.51 | -0.24 | 0.02 | 0.29 | 0.55 | 0.82 | 1.08 |
| 8 | -1.19 | -0.96 | -0.73 | -0.50 | -0.27 | -0.04 | 0.19 | 0.42 | 0.65 | 0.88 | 1.11 |
| 9 | -0.90 | -0.69 | -0.49 | -0.29 | -0.08 | 0.12 | 0.32 | 0.53 | 0.73 | 0.93 | 1.14 |
| 10 | -0.66 | -0.48 | -0.30 | -0.11 | 0.07 | 0.25 | 0.43 | 0.61 | 0.79 | 0.98 | 1.16 |
| 15 | 0.08 | 0.19 | 0.31 | 0.43 | 0.54 | 0.66 | 0.77 | 0.89 | 1.01 | 1.12 | 1.24 |
| 20 | 0.48 | 0.56 | 0.64 | 0.72 | 0.81 | 0.89 | 0.97 | 1.05 | 1.13 | 1.21 | 1.30 |

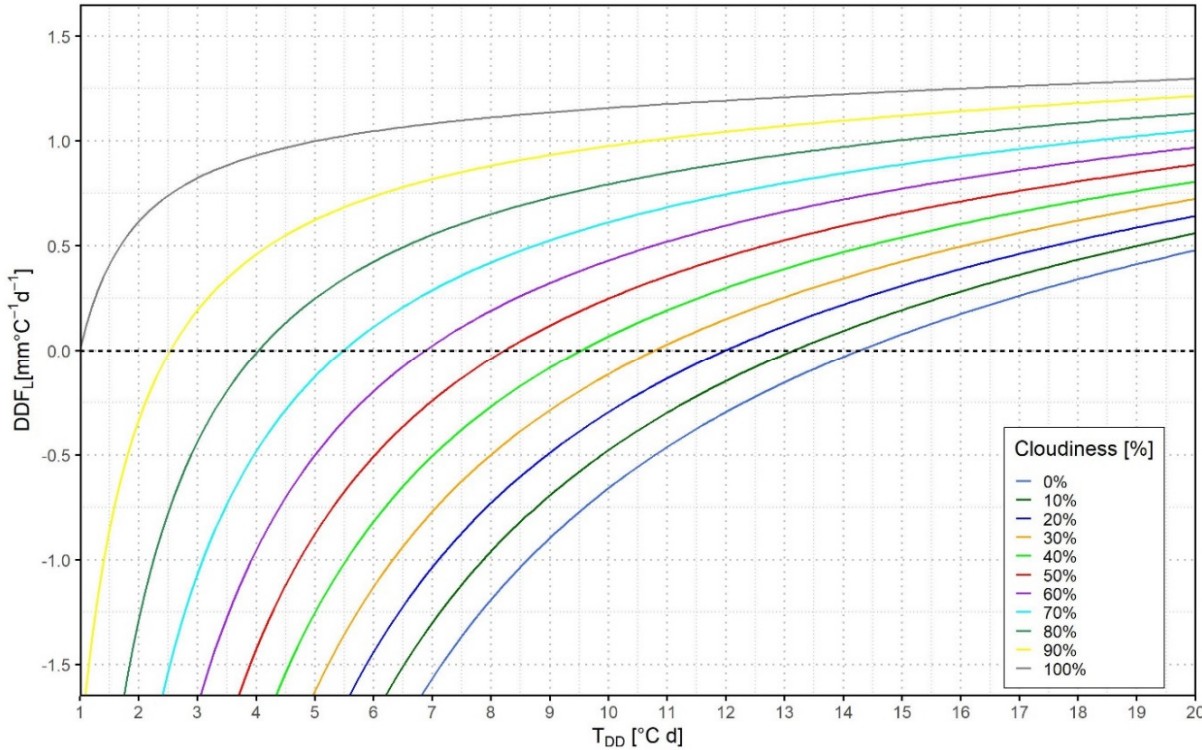

**Figure 6** Longwave Radiation component ($DDF_L$) for selected cloudiness [%] and degree-days [°C d]

### 4.3 Sensible heat component - $DDF_H$

The sensible heat flux $Q_H$ as given by eq. (26) is mainly proportional to wind speed and the temperature difference between air and snow surface. Furthermore, air density, besides its dependency on temperature, is a function of relative humidity and atmospheric pressure, and thus of altitude (eq. (27) and (28)). Since the influence of the relative humidity on air density is negligible, a relative humidity of $RH = 0\%$ is assumed in the below analysis on the response of the $DDF_H$ to changes in temperature resp. degree-days, wind speed and altitude. It should be noted that this analysis assumes typical melt conditions with a snowpack temperature of $T_s = 0$ °C and positive air temperature, whereas negative air temperature would lead to a negative sensible heat flux resulting in a cooling of the snowpack and a decrease of total $DDF$.

Table 3 presents the variation in $DDF_H$ depending on wind speed and degree-days, while altitude is assumed constant at sea level. Results in Table 3 show that there is only a minor effect on $DDF_H$ values due to increase in degree-days compared to the influence of wind speed. For example, for a daily average wind speed of $u = 1.0$ m s$^{-1}$, the $DDF_H$ only decreases from 0.806 to 0.781 mm °C$^{-1}$ d$^{-1}$ when degree-days increase from 1 to 10 °C d. On the other hand, for a degree-day of 1 °C d, the $DDF_H$ increases proportionally from 0.806 to 8.061 mm °C$^{-1}$ d$^{-1}$ when wind speed increases from 1 to 10 m s$^{-1}$. Thus, wind speed is a decisive variable when estimating the $DDF_H$.

Table 4 and Figure 7 show the variation in $DDF_H$ depending on altitude and degree-days, while the wind speed is assumed to be constant at $u = 1$ m s$^{-1}$. The latter allows the $DDF_H$ to be easily calculated for any other wind speed by multiplying the given value by the actual wind speed. The $DDF_H$ principally decreases with altitude, with less pronounced differences due to temperature at higher altitudes.

If wind speed observations are not available, they may be roughly estimated based on the topographic and climate characteristics of the study area. Stigter et al., (2021) for example give a range of wind speed at two different sites in the central Himalayas. At Ganja La the wind speed is generally low i.e. < 2 m s$^{-1}$ and has no distinct diurnal cycle, whereas at Yala the wind speed exhibit a strong diurnal cycle with wind speeds ≥ 5 m s$^{-1}$ occurring in the afternoon during the entire snow season. Dadic et al., (2013) found values around 3 – 5 m s$^{-1}$ for a glaciered catchment in Switzerland. However, average values may not represent the actual wind conditions and thus $DDF_H$ on a certain day. While for example the geometric mean of observed daily wind speed at the Brunnenkopfhütte station is about 0.8 m s$^{-1}$ resulting in a $DDF_H$ of approx. 0.7 mm °C$^{-1}$ d$^{-1}$, the maximum daily average wind speed is about 4.5 m s$^{-1}$ which increases $DDF_H$ to approx. 3.9 mm °C$^{-1}$ d$^{-1}$.

**Table 3**    Sensible heat component ($DDF_H$) [mm °C$^{-1}$ d$^{-1}$] for selected wind speed [m s$^{-1}$] and degree-days [°C d]

| | $DDF_H$ | | | | | | | |
| --- | --- | --- | --- | --- | --- | --- | --- | --- |
| | Wind Speed [m s$^{-1}$] | | | | | | | |
| $T_{DD}$ [°C d] | 0.1 | 0.5 | 1 | 2 | 3 | 4 | 5 | 10 |
| 1 | 0.081 | 0.403 | 0.806 | 1.612 | 2.418 | 3.225 | 4.031 | 8.061 |
| 5 | 0.079 | 0.397 | 0.795 | 1.589 | 2.384 | 3.178 | 3.973 | 7.945 |
| 10 | 0.078 | 0.390 | 0.781 | 1.561 | 2.342 | 3.122 | 3.903 | 7.805 |
| 15 | 0.077 | 0.383 | 0.767 | 1.534 | 2.301 | 3.068 | 3.835 | 7.670 |
| 20 | 0.075 | 0.377 | 0.754 | 1.508 | 2.262 | 3.016 | 3.769 | 7.539 |

*Note: Air density values are assumed at an elevation of 0 m a.s.l. and RH=0%.*

**Table 4**    Sensible heat component ($DDF_H$) [mm °C$^{-1}$ d$^{-1}$] for selected altitude [m a.s.l.] and degree-days [°C d]

| | $DDF_H$ | | | | | |
| --- | --- | --- | --- | --- | --- | --- |
| | Altitudes [m a.s.l.] | | | | | |
| $T_{DD}$ [°C d] | 0 | 1000 | 2000 | 3000 | 4000 | 5000 |
| 1 | 0.806 | 0.712 | 0.628 | 0.555 | 0.490 | 0.432 |
| 5 | 0.795 | 0.703 | 0.621 | 0.550 | 0.486 | 0.430 |
| 10 | 0.781 | 0.692 | 0.613 | 0.543 | 0.482 | 0.427 |
| 15 | 0.767 | 0.681 | 0.605 | 0.537 | 0.477 | 0.424 |
| 20 | 0.754 | 0.671 | 0.597 | 0.531 | 0.473 | 0.421 |

*Note: RH= 0%, u = 1.0 m s$^{-1}$*

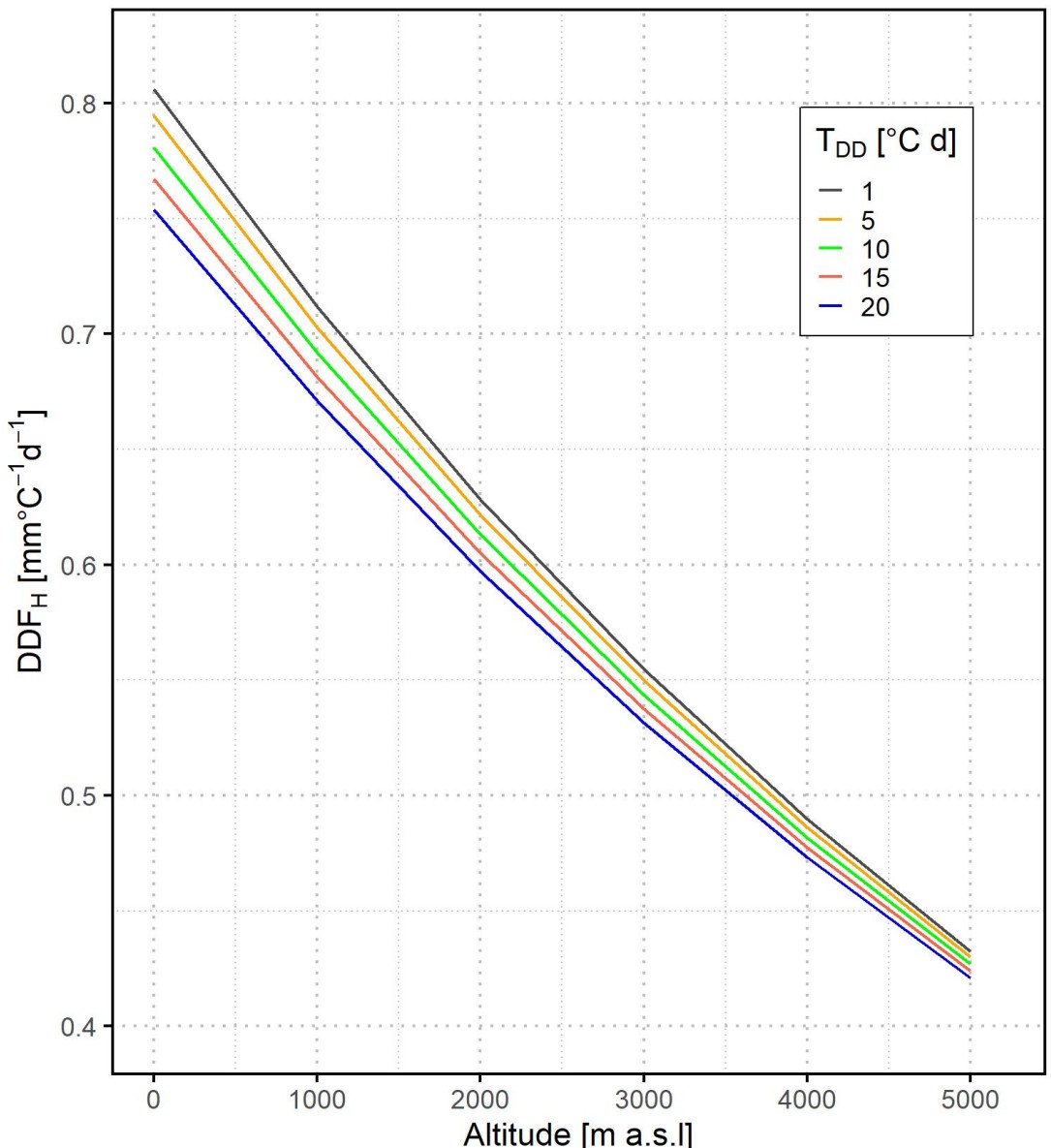

**Figure 7**     Variation of sensible heat component (DDF$_H$) at different altitude based on different degree-days (note: $RH = 0\%$ $u = 1$ m s$^{-1}$)

### 4.4 Latent heat component - DDF$_E$

The latent heat flux $Q_E$ approximated by an aerodynamic model as in eq. (30) shows, that the latent heat component DDF$_E$ is mainly dependent on the humidity gradient near the snow surface and on the wind speed. Additionally, altitude has an influence, as the air density is decreasing with altitude. Table 5 and Figure 8 give the resulting DDF$_E$ as a function of degree-days for different values of relative humidity and at daily average wind of $u = 1.0$ m s$^{-1}$ whereas air density values are assumed at an elevation of 0 m a.s.l. In line with the sensible heat component DDF$_H$, DDF$_E$ for any other wind speed can be obtained by multiplication by the actual value. For relative humidity < 30% the DDF$_E$ is negative over the whole range of degree-days, hence the latent heat component will reduce the total *DDF* under these conditions. Even if the air is humid and warm, contribution of latent heat is moderate, e.g. DDF$_E$ = 1.0 mm °C$^{-1}$ d$^{-1}$ at a relative humidity of 100% and $T_{DD} = 20$ °C d.

Figure 9 shows the combined effect of altitude, relative humidity, and temperature on $DDF_E$. At a high relative humidity (e.g. $RH = 100\%$), similar to the $DDF_H$ the $DDF_E$ values principally decrease with altitude, with less pronounced differences due to temperature at higher altitudes. At lower relative humidity (e.g. $RH = 50\%$), the altitude effect is less noticeable and at low temperatures even a reversal of the effect can be observed. Thus, altitude reduces positive $DDF_E$ associated with high humidity while it also reduces the cooling effect of a negative latent heat flux, which is associated with low humidity and lower air temperature.

As the above analysis shows, humidity is the main variable influencing the $DDF_E$. In general, humid air will promote condensation at a cooler snow surface, which releases latent energy and contributes to a positive $DDF$, while dry air will promote evaporation and sublimation from the snow surface, which abstracts energy from the snowpack. Thus, mainly depending on the humidity of the air, the latent heat energy flux is usually a heat sink while only in case of high humidity in conjunction with higher temperature it becomes a heat source to the snowpack. Especially in spring, when relative humidity is comparatively low in middle and northern latitudes, large parts of the incoming solar radiation can be consumed by evaporation from the snow surface reducing significantly the energy available for melt and thus reducing the corresponding $DDFs$ (Lang and Braun, 1990; Zhang et al., 2006).

**Table 5**  Latent heat component ($DDF_E$) [mm °C$^{-1}$ d$^{-1}$] for selected relative humidity [%], degree-days [°C d] and wind speed $u = 1$ [m s$^{-1}$]

| | DDF$_E$ | | | | | | | | | |
|---|---|---|---|---|---|---|---|---|---|---|
| | Relative Humidity | | | | | | | | | |
| $T_{DD}$ [°C d] | 10% | 20% | 30% | 40% | 50% | 60% | 70% | 80% | 90% | 100% |
| 1 | -6.91 | -6.19 | -5.42 | -4.62 | -3.79 | -2.94 | -2.07 | -1.18 | -0.28 | 0.57 |
| 2 | -3.42 | -3.03 | -2.61 | -2.18 | -1.73 | -1.27 | -0.80 | -0.32 | 0.15 | 0.58 |
| 3 | -2.25 | -1.97 | -1.67 | -1.36 | -1.04 | -0.71 | -0.37 | -0.02 | 0.29 | 0.60 |
| 4 | -1.67 | -1.44 | -1.20 | -0.95 | -0.69 | -0.42 | -0.14 | 0.12 | 0.37 | 0.62 |
| 5 | -1.32 | -1.12 | -0.91 | -0.70 | -0.47 | -0.24 | 0.00 | 0.21 | 0.42 | 0.64 |
| 6 | -1.08 | -0.91 | -0.72 | -0.52 | -0.32 | -0.11 | 0.09 | 0.28 | 0.47 | 0.65 |
| 7 | -0.91 | -0.75 | -0.58 | -0.40 | -0.21 | -0.02 | 0.16 | 0.33 | 0.50 | 0.67 |
| 8 | -0.79 | -0.63 | -0.47 | -0.30 | -0.13 | 0.05 | 0.21 | 0.37 | 0.53 | 0.69 |
| 9 | -0.69 | -0.54 | -0.39 | -0.22 | -0.06 | 0.10 | 0.26 | 0.41 | 0.56 | 0.72 |
| 10 | -0.61 | -0.47 | -0.32 | -0.16 | 0.00 | 0.15 | 0.30 | 0.44 | 0.59 | 0.74 |
| 15 | -0.36 | -0.23 | -0.09 | 0.06 | 0.19 | 0.32 | 0.46 | 0.59 | 0.72 | 0.86 |
| 20 | -0.23 | -0.09 | 0.05 | 0.19 | 0.32 | 0.46 | 0.59 | 0.73 | 0.86 | 1.00 |

*Note: These values are for $u=1$ m s$^{-1}$, for a different wind speed these values can be multiplied for desired wind speed. Air density values are assumed at an elevation of 0 m a.s.l.*

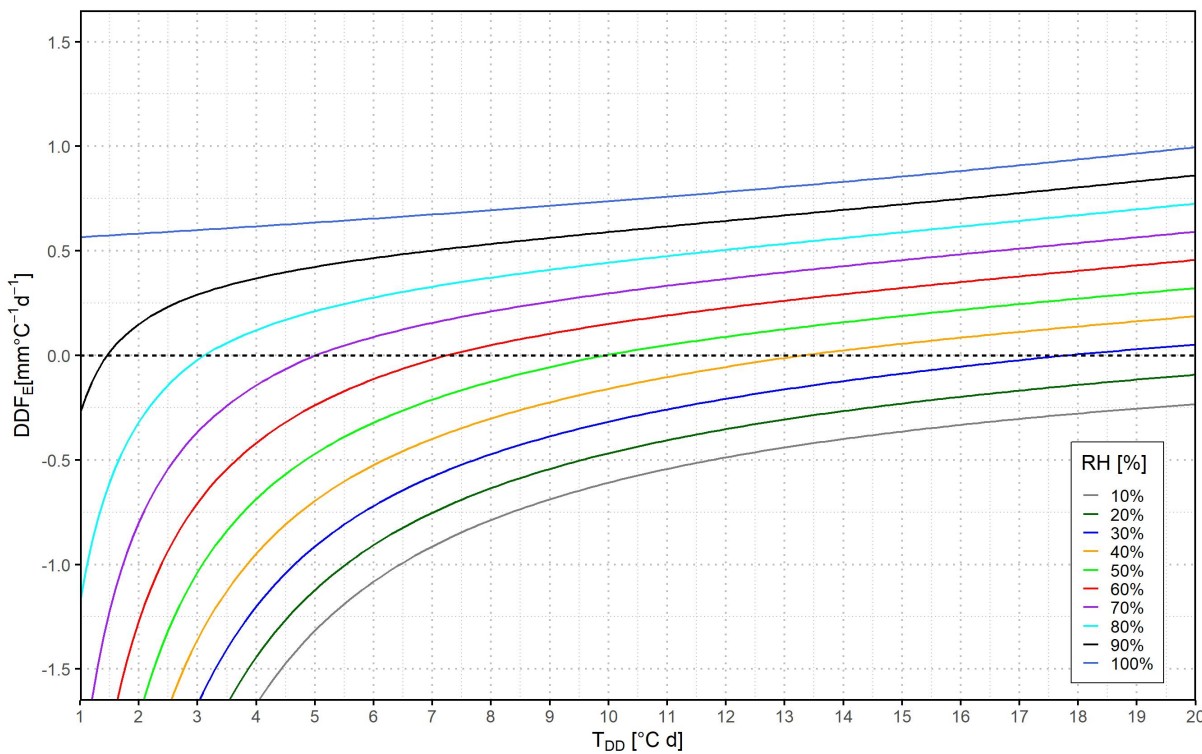

**Figure 8** Latent Heat component (DDF$_E$) for selected relative humidity [%], degree-days [°C d] and $u = 1$ [m s$^{-1}$]

*Note: These values are for u=1 m s$^{-1}$, for a different wind speed these values can be multiplied for desired wind speed. Air density values are assumed at an elevation of 0 m a.s.l.*

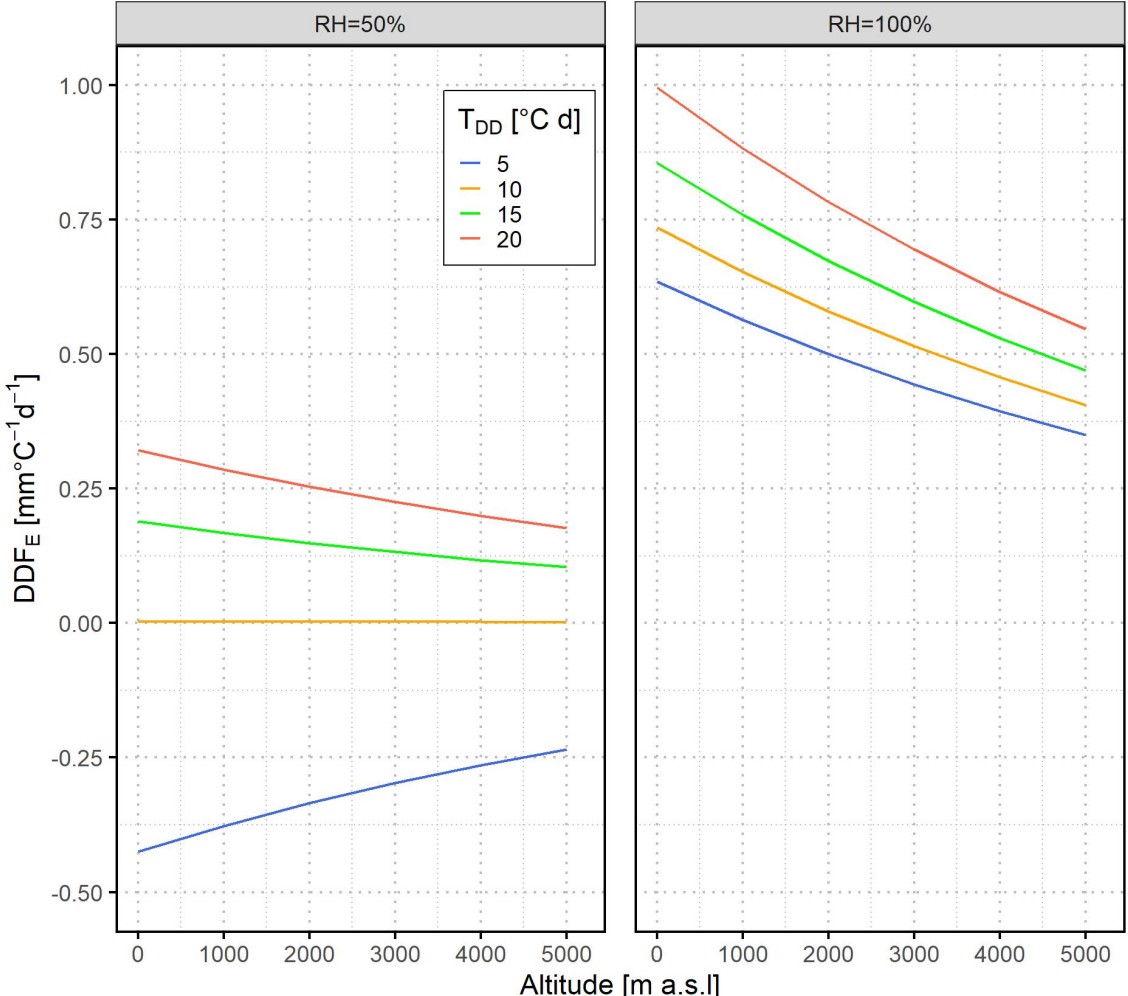

**Figure 9**    Variation of latent heat component (DDF$_E$) depending on altitude for different relative humidity values and degree-days (note: $u$ =1 m s$^{-1}$)

### 4.5 Precipitation heat component – DDF$_P$

Rainfall can affect the snowpack energy budget by adding sensible heat due to warm rain and by release of latent heat if the rain refreezes in the snowpack (DeWalle and Rango, 2008). The latter effect is not considered in this study, as the snowpack is assumed at 0°C melting condition. Because according to eq. (34) the precipitation heat $Q_P$ is linearly dependent on air temperature, division by respective degree-days makes DDF$_P$ independent of temperature and proportional to rainfall, resulting in a DDF$_P$ = 0.0125 mm °C$^{-1}$ d$^{-1}$ for a precipitation depth of 1 mm per day. DDF$_P$ for any other precipitation can be obtained by respective multiplication. The exemplary values in Table 6 show however, that the contribution of precipitation heat component DDF$_P$ is modest compared to other DDF components. Even high rainfall of 50 mm in a day would release only a small amount of sensible heat, resulting in a DDF$_P$ of 0.6 mm °C$^{-1}$ d$^{-1}$.

**Table 6**    Precipitation heat component (DDF$_P$) [mm °C$^{-1}$ d$^{-1}$] for selected precipitation [mm d$^{-1}$]

| Precipitation ($P$) | 1 | 2 | 5 | 10 | 25 | 50 |
|---|---|---|---|---|---|---|
| DDF$_P$ | 0.0125 | 0.025 | 0.0625 | 0.125 | 0.313 | 0.625 |

**5. Discussion**

While the previous section focuses on the characteristic of each individual energy flux based *DDF* component, this section mainly discusses the influence of spatial, seasonal or meteorological conditions on the overall *DDF*. The discussion section bifurcates into two sub-sections, (i) influence of individual factors on the *DDF* such as latitude, altitude, albedo, season and rain on snow events, and (ii) application of energy flux based *DDF* estimates, which shows how energy flux based *DDF* can be estimated for a temperature-index model by using different available datasets and applied under varying conditions, e.g. meteorological and climate change conditions.

**5.1 Influence of individual factors on the DDF**

In this section all conclusions are under the assumption that the snowpack is isothermal at $T_s = 0$ °C and in ripe condition, hence all net incoming energy is available for melt and contributes to the total *DDF*. Other than the discussed variables are assumed constant with standard values $u = 1$ m s$^{-1}$, $RH = 70\%$, $A = 0.5$, $P = 0$ mm, and typical melt conditions of $T_{DD} = 5$ °C d if not stated otherwise.

**5.1.1 Influence of latitude**

While topographic factors like slope, aspect or shading in mountainous regions result in a high local variability of melt conditions, larger scale regional pattern of *DDFs* like a dependency on latitude could not be detected in a data review by Hock, (2003). This observation is supported by a brief analysis of the effect of latitude below, where the *DDF* is compared not on the same date but at same degree-days. As an illustrative example, typical melt conditions of $T_{DD} = 5$ °C d at a latitude of about 35° North in the Upper Jhelum catchment (Bogacki and Ismail, 2016) are compared to similar conditions at a latitude of 48° North (Brunnenkopfhütte, 1602 m a.s.l.). As zone-wise temperature data (see Sec. 2.2) indicates, in the Upper Jhelum catchment at an elevation zone of 1500 – 2000 m a.s.l. above melting conditions usually occur around mid-February while at Brunnenkopfhütte comparable degree-days arrive about one month later in mid-March. Figure 10 (a) compares the energy flux based *DDF* components at both latitudes. The decisive solar radiation component is very similar at the two locations, both under clear sky and overcast conditions, thus the total *DDF* is virtually identical at both latitudes. Therefore, at least in moderate latitudes and when compared under similar melt conditions, no significant effect of latitude on *DDF* could be found.

**5.1.2 Influence of altitude**

Contrary to the compensating effect in the case of latitude, the delayed onset of snowmelt due to altitude influences the *DDF* noticeably, which becomes important in temperature-index models where calculation is usually based on elevation bands. In order to demonstrate the influence of altitude on the *DDF*, two elevation zones with an altitude of 1500 – 2000 and 3500 – 4000 m a.s.l. respectively are compared at 35° latitude in the Upper Jhelum catchment. As already mentioned, typical melt conditions

of $T_{DD}$ = 5 °C d occur at 1500 – 2000 m a.s.l. usually around mid-February, while at 3500 – 4000 m a.s.l. similar degree-days arrive about mid-May. The resulting *DDFs* (see Figure 10 (b)) show a significant difference, both under clear sky as under overcast conditions, because of the different input in solar radiation caused by the alteration in solar angle between February and May. Figure 10 (b) shows an additional term DDF$_A$ on top of the solar radiation component that represents the increase in incoming solar radiation due to the clearness altitude factor, which takes into account the increase of the clearness index with altitude. Averaging the factors proposed by different solar radiation models (see Figure 4) results in an additional component DDF$_A$ of 0.4 and 1.4 mm °C$^{-1}$ d$^{-1}$ under clear sky and of 0.5 and 1.6 mm °C$^{-1}$ d$^{-1}$ und overcast conditions at 1500 – 2000 and 3500 – 4000 m a.s.l. respectively.

While for exemplification snow albedo is assumed constant at 0.5 in Figure 10 (b), taking into consideration the decrease of albedo as the snow ages (see Table 1) e.g. $A$ = 0.74 in February and $A$ = 0.42 in May results in a more pronounced difference with altitude, i.e. a total *DDF* of 0.3 compared to 10.5 mm °C$^{-1}$ d$^{-1}$ under clear sky and of 2.7 versus 7.3 mm °C$^{-1}$ d$^{-1}$ under overcast conditions for the two altitudes respectively.

The increase of *DDF* with increasing altitude has already been mentioned in previous studies (e.g. (Hock, 2003; Kayastha and Kayastha, 2020)). The *DDF* estimates presented in this study are in line with previous studies. For example, in Nepalese Himalayan region, seasonal average *DDF* increases with respect to altitude from 7.7 – 11.6 mm d$^{-1}$ °C$^{-1}$ (Kayastha et al., 2000) whereas Kayastha and Kayastha, (2020) found model calibrated range of the *DDF* in central Himalayan basin is between 7.0 – 9.0 mm d$^{-1}$ °C$^{-1}$. As Kayastha et al., (2000) pointed out higher values of the *DDF* usually occur at very low temperatures at higher altitudes which refers as 'the low temperature effect' (Kayastha et al., 2000), since at higher altitudes the major driving factor to melt is the energy input by solar radiation.

**5.1.3 Influence of albedo**

As already discussed in the sections before, snow albedo is a critical parameter for the *DDF* since according eq. (5) albedo directly controls the net solar radiation flux into the snowpack. While albedo of fresh snow is well above 0.9 hence reflecting most of the incoming shortwave radiation, it drops rapidly when larger grains form due to snow metamorphism. Figure 10 (c) demonstrates the effect of aging snow after a new snow event, when a simple exponential decay model as given in eq. (19) is used and typical melting conditions $T_{DD}$ = 5 °C d are assumed. Since directly after a new snow event (Day = 0) the fresh snow albedo is high ($A$ = 0.95), the overall *DDF* is generally small. Under clear sky conditions, in case longwave radiation cooling is larger than net shortwave radiation flux, even a negative *DDF* value, i.e. no melt, may occur. If there is no new snow event in-between, albedo will decrease following the exponential decay model to 0.52 after 10 days resulting in a *DDF* of 5.8 mm °C$^{-1}$ d$^{-1}$ under clear sky and 4.4 mm °C$^{-1}$ d$^{-1}$ under overcast conditions. The increase in the *DDF* with exponential decay in albedo is in agreement with the findings of MacDougall et al., (2011), who found that the *DDF* is sensitive to albedo with values of > 4.0 mm d$^{-1}$ °C$^{-1}$ at an albedo on 0.6. As described

qualitatively in the literature e.g. (Hock, 2003), under all sky conditions the *DDF* is continuously increasing with decreasing albedo, with the increase however being more pronounced under clear sky than under overcast conditions.

**5.1.4 Influence of season**

Since the solar angle is rising from its minimum at winter solstice in December to its maximum on 21$^{st}$ June, the solar radiation component DDF$_S$ is increasing during the snowmelt season and thus the *DDF* is expected to increase respectively. Figure 10 (d) shows the influence of season on the *DDF* at the Brunnenkopfhütte test site during the melt period, assuming average degree-days of 1, 4, and 7 °C d in March, April, and May respectively (see Table 1). Under clear sky conditions, as expected total *DDF* increases from a negative value of -3.6 mm °C$^{-1}$ d$^{-1}$ in March to 6.6 mm °C$^{-1}$ d$^{-1}$ in May. Under overcast conditions however, the *DDF* is virtually stable ranging from 4.4 to 4.5 mm °C$^{-1}$ d$^{-1}$ in the same period. The stability of the *DDF* under overcast conditions found in the present study is in agreement with the study of Kayastha et al., (2000), who found that the *DDF* observed during July – August are small compared to June because of prevailing cloud cover due to monsoon activity which reduces the incoming shortwave radiation.

An evaluation of the individual *DDF* components shows, that under clear sky conditions the high impact of solar radiation in combination with low degree-days at the onset of the snowmelt season is counterweighted by a strong negative longwave radiation component that decreases as the season progresses. Under overcast conditions, DDF$_L$ is neutral or slightly positive while the DDF$_S$ component decreases because degree-days are rising faster than solar radiation input, which implies that sky conditions are more decisive for an estimate of the *DDF* than the date.

The effect of cloud cover is further amplified by the decrease in albedo while the melt season progresses, which becomes more significant under clear sky conditions. In the present example, that uses the average monthly albedo as specified in Table 1, only 30% of incoming solar radiation is contributing to melt in March, while it is about 60% in May, enhancing the marked increase of the *DDF* under clear sky conditions.

**5.1.5 Influence of rain on snow events**

In general, precipitation heat component alone has only a minor effect on the *DDF*. However, in conjunction with certain weather conditions like breaking in of warm and moist air, rain over snow events may lead to sudden melt and severe flooding. In a well-documented event in the Alps in October 2011 (Rössler et al., 2014) intensive rainfall (on average 100 mm d$^{-1}$) was accompanied by an increase in temperature by 9 °C, which shifted the 0 °C line from 1500 to 3200 m a.s.l. during one day. Similar conditions occurred e.g. end-December 2021 in Switzerland, when after the establishment of a solid snow cover, an Atlantic cyclone caused a sudden temperature rise up to 19 °C, wind gusts of 35 – 40 m

s$^{-1}$ and locally more than 70 mm precipitation (MeteoSchweiz, 2022), which e.g. at Adelboden (1325 m a.s.l.) caused the complete melt of an approx. 40 cm snow cover.

Figure 10 (e) shows the different *DDF* components resulting from a hypothetical rain over snow event assuming an air temperature of 15 °C, a precipitation of 70 mm d$^{-1}$, a daily average wind speed of 10 m s$^{-1}$, a relative humidity of 100%, and overcast conditions. Although the amount of precipitation is substantial and rain's temperature is comparatively high, the contribution of DDF$_P$ is still modest. However, air temperature, relative humidity, and in particular wind speed associated with such events increase the sensible and latent heat components significantly. Thus, the resulting overall *DDF* is much higher than under usual melt conditions, which may lead to a considerable melt that adds to the runoff already caused by the heavy rain.

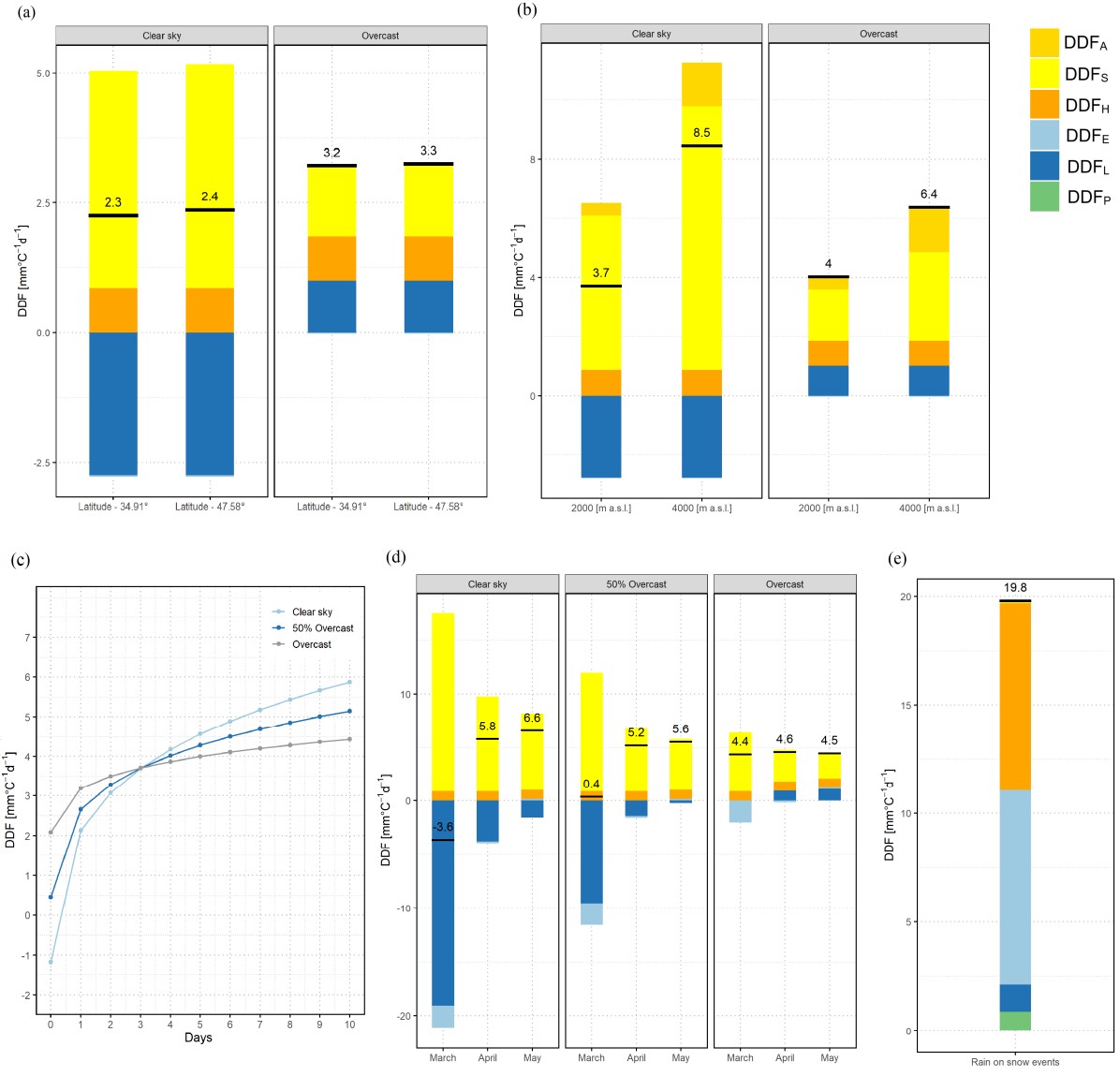

Figure 10   Influence of (a) latitude; (b) altitude; (c) albedo; (d) season; (e) rain on snow events – on the *DDF* under clear sky and overcast conditions

**5.2 Application of energy flux based DDF estimates**

 **5.2.1 DDF estimates under field conditions**

In addition to the analysis of the influence of individual factors on the *DDF*, the dataset from the Brunnenkopfhütte test site is used to compare energy flux based estimates with field-derived DDFs in order to demonstrate how naturally varying meteorological conditions during the melt season, and in particular the cold content of the snowpack, affect the accuracy of *DDF* estimates.

For this purpose, daily melt was estimated from the daily difference of observed snow water equivalent during melt periods (see Figure 3). Energy flux based melt was calculated by the formulas given in Sec. 3 using observed daily data from the Brunnenkopfhütte automatic snow and weather station (e.g. air temperature, wind speed, etc., see Sec. 2.1) where applicable.

The daily degree-day sum is calculated from hourly air temperature data as proposed by Braithwaite and Hughes (2022). In general, operational degree-day models typically use constant degree-day factors for a certain time period (e.g. 10-daily period). In this backdrop, both energy flux based and data-derived daily melt values were accumulated on 10-daily basis and divided by the degree-days of the respective period. The 10-daily averaging procedure also smooths daily noise in the observed data, in particular inaccuracies in the determination of daily melt and unrealistic *DDF* values because of daily temperature averages just above 0 °C.

The comparison between field-derived and estimated (energy flux based) *DDFs* (see Figure 11) yields a fair agreement with bias = 0.14 mm °C$^{-1}$ d$^{-1}$ between estimated and field-derived values, and Root Mean Square Error (RMSE) = 1.12 mm °C$^{-1}$ d$^{-1}$. Noteworthy in Figure 11 are the 10-daily melt periods marked by hollow circles that were excluded from the calculation of the error metrics as new snow events occurred during these periods. Due to fresh snow, the snowpack is no longer 'ripe' and a certain amount of the incoming energy is needed to bring it back to that state, thus does not contribute to melt. This effect can be clearly seen in Figure 11, since all estimated *DDFs* belonging to these periods considerably overestimate the field-derived ones.

Certainly it is of interest to estimate *DDFs* also in cases where the snowpack, e.g. because of new snow events or due to radiational cooling during clear cold nights, is not 'ripe'. An approach to account for the snowpack's energy deficit, i.e. the energy needed to bring the snowpack temperature isothermal at 0°C, is the concept of 'cold content' (Marks et al., 1999; Schaefli and Huss, 2011). The cold content is usually either estimated as a function of meteorological parameters or calculated by keeping track of the residuals of the snowpack energy balance (Jennings et al., 2018). For the latter, the SNOWPACK model (Lehning et al., 2002) is an excellent tool, which provides a highly detailed simulation of the vertical mass, energy, and besides other state variables the snow temperature distribution inside a snowpack. However, SNOWPACK requires a considerable number of meteorological input variables and

preferably at least hourly observations, both of which are usually not available in the context where degree-day models are employed.

Especially suited for data scarce conditions, Walter et al., (2005) apply a lumped approach, that accounts for the cold content by changing the (isothermal) snowpack temperature depending on the daily net energy flux. When the incoming energy flux is sufficient to raise the snow temperature to 0°C or when it is already at 0°C the day before, all additional available energy produces melt. This appealing approach, which does not need any additional data, however seems to significantly over-estimate the

snowpack temperature in particular in situations with negative energy fluxes at night but a positive daily net balance, as a comparison with SNOWPACK simulations and data from Brunnenkopfhütte shows. Therefore, an appropriate parametrisation of the cold content under limited data availability that would enable satisfactory estimates of *DDFs* in situations when the snowpack is not completely ripe, remains subject to further research.

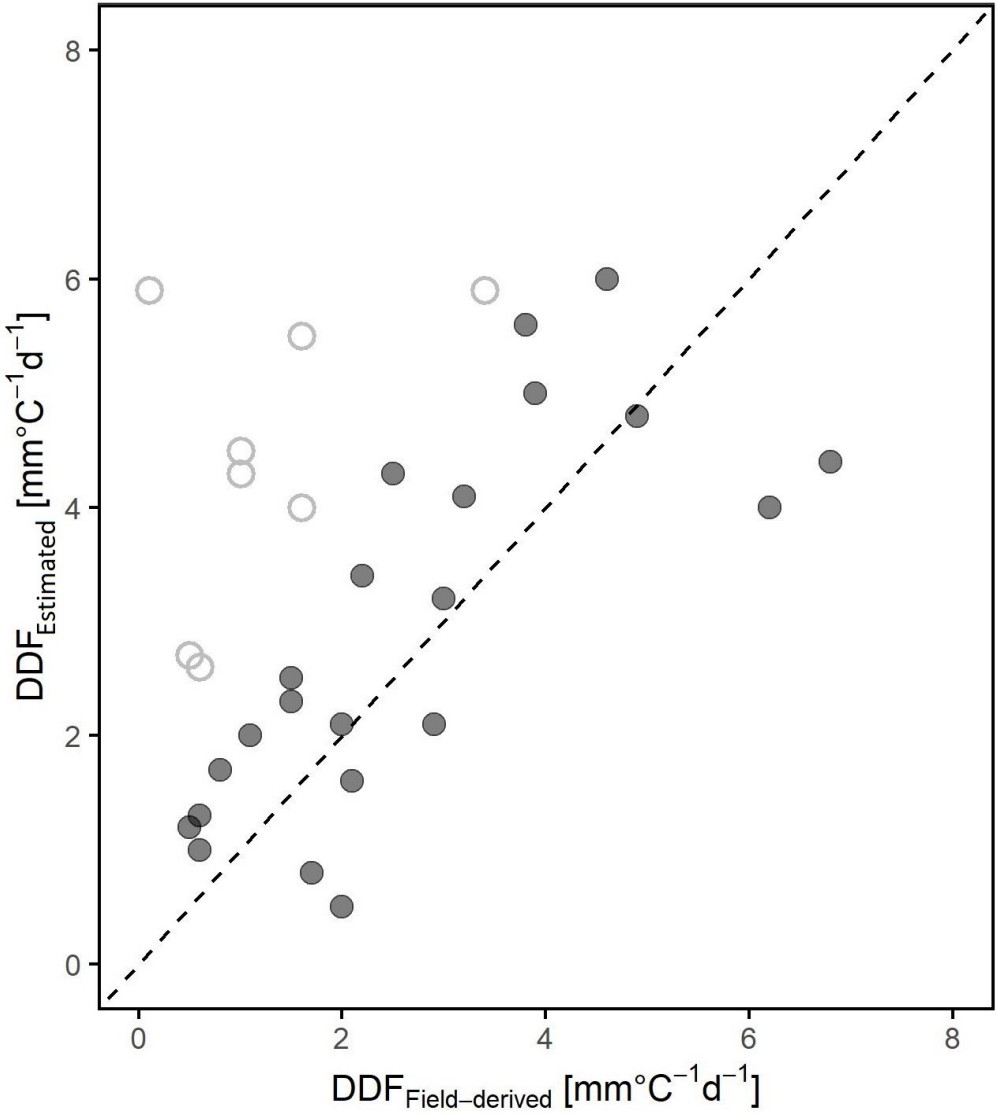

Data source: Brunnenkopfhütte - [Period: 2016/17-2020/21]

**Figure 11** Comparison of field-derived vs estimated (energy flux based) 10-daily *DDF* for the Brunnenkopfhütte test site (period: November 2016 – May 2021) – Hollow points represent *DDFs* during periods with new snow events

### 5.2.2 DDF estimates for temperature-index modelling

Snowmelt runoff models using the temperature-index approach have proven useful tools for simulation and forecasting in large snow or glacier dominated catchments, in particular in remote mountainous regions where data is usually scarce. A good estimate of the degree-day factor as the decisive model parameter is important either to stay in a realistic range when calibrating this parameter or in case of forecasting when estimating its changes while the season progresses. In order to demonstrate the alteration of *DDFs* over time and altitude, energy flux based *DDFs* are estimated using 10-daily average temperature (i.e. period 2000 – 2015) for the key elevation zones in the Upper Jhelum catchment (Bogacki and Ismail, 2016). Because of the lack of other than temperature and precipitation data,

prevailing conditions during the melt season are crudely approximated by the standard conditions used in this section, assuming persistent clear sky conditions and albedo declining according eq. (19) after last fresh snow just before the beginning of the melting period.

Figure 12 (a) shows the development of *DDFs* in the elevation zones over time. As expected, melt starts earlier in lower elevation zones and successively progresses to higher altitudes. Interestingly, the *DDF* in the first 10-daily period of melting in each elevation zone increases with altitude. Obviously this is a combined effect of higher solar radiation input and decreasing albedo while the season progresses and the circumstance that the onset of melt in higher elevation zones starts at a lower degree-day threshold than in lower zones. In contrast to Figure 10 (d), the *DDF* in Figure 12 decreases continuously in all elevation zones in the subsequent melting periods since air temperature and thus degree-days rise faster than melt. The range of *DDFs* for snow estimated by the energy flux components is in good agreement with earlier studies for the Himalayan region, e.g. 7.7 – 11.6 mm $d^{-1}$ $°C^{-1}$ (Kayastha et al., 2000), 5 – 9 mm $d^{-1}$ $°C^{-1}$ (Zhang et al., 2006), 5 – 7 mm $d^{-1}$ $°C^{-1}$ (Tahir et al., 2011) and 7.0 – 9.0 mm $d^{-1}$ $°C^{-1}$ (Kayastha and Kayastha, 2020).

**5.2.3 DDF estimates under the influence of climate change**

Climate change will ultimately influence snowmelt patterns depending on the projected changes in temperature and precipitation. In recent studies, usually model parameters including *DDFs* are considered as constant when assessing the climate change impact on future water availability from snow and glacier fed catchments (Lutz et al., 2016; Hasson et al., 2019; Ismail et al., 2020). However, due to the physical processes on which they depend these parameters are subject to climate change. In this section, an attempt is made to estimate the influence of climate change on the *DDFs* in different elevation zones. For this analysis, results from ISIMIP data (see Sec. 2.2), which predict the temperature change for the period 2071 – 2100 to $\Delta T$ = 2.3 °C under RCP2.6 and $\Delta T$ = 6.5 °C under RCP8.5, are added to the temperatures in present climate for each elevation zone.

The first effect to be observed in Figure 12 (b) and (c) is the common finding that snowmelt will start earlier under climate change as temperatures rise earlier above freezing. In addition, since being earlier in the year, the *DDFs* in corresponding elevation zones are generally smaller compared to the current climate, though there are some outliers at the start of melting, due to division by low degree-day values. In case of the pessimistic scenario RCP8.5 (Figure 12 (c)), a seasonal snow cover will not establish any more in the lowest elevation zone (i.e. 2500 – 3000 m a.s.l.) as air temperature at this elevation is projected to stay well above freezing throughout the winter. In general, the results of this brief analysis indicate, that the *DDFs* are expected to decrease under the influence of climate change, as snowmelt season will shift earlier in the year when solar radiation is small and snow albedo values are expected to be on higher side. Musselman et al., (2017), highlight similar findings about slower snowmelt in a warmer world due to a shift of the snowmelt season to a time of lower available energy.

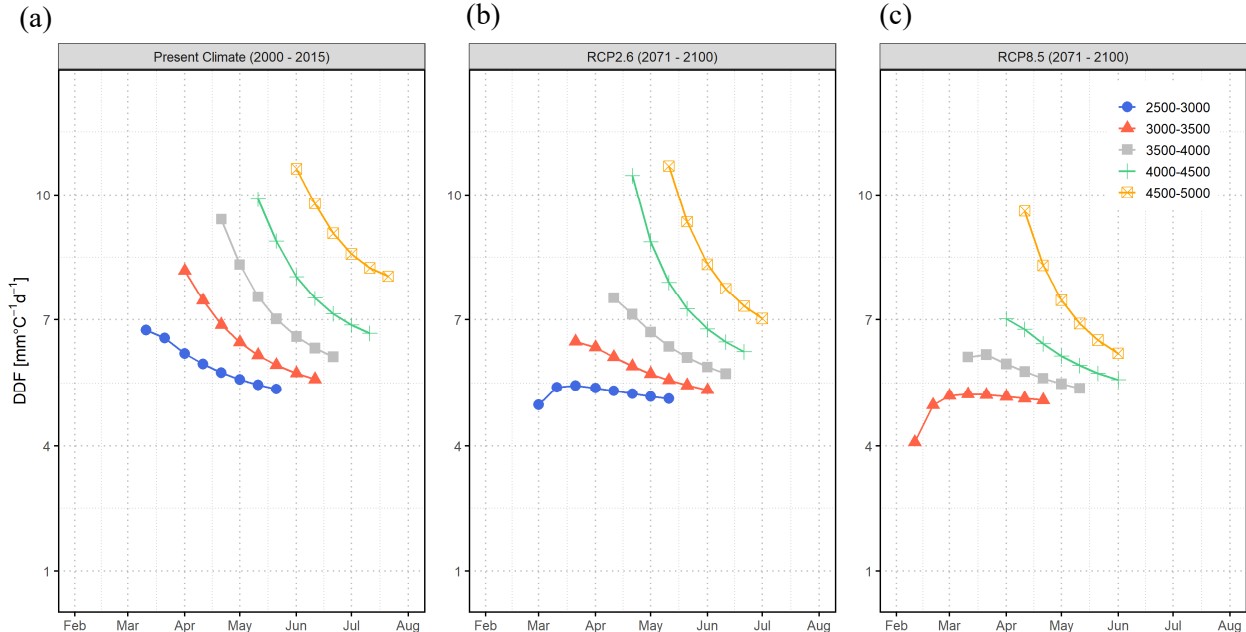

**Figure 12** (a) *DDF* estimates for a temperature-index modelling in present climate; (b) Influence of climate change – 2071 – 2100 under RCP2.6; (c) Influence of climate change – 2071 – 2100 under RCP8.5

## 6. Conclusions

Degree-day models are common and valuable tools for assessing present and future water availability in large snow or glacier melt dominated basins, in particular when data is scarce like e.g. in the Hindukush-Karakoram-Himalayas mountain ranges. The present study attempts to quantify the effects of spatial, temporal, and climatic conditions on the degree-day factor (*DDF*), in order to gain a better understanding which influencing factors are decisive under which conditions. While this analysis is physically based on the energy balance, formulas with minimum data requirement for estimating the *DDFs* are used to account for situations where observed data is limited. In addition, resulting tables and graphs for typical melt conditions are provided for a quick assessment.

A comparison between field-derived and estimated *DDFs* at the Brunnenkopfhütte test site shows a fair agreement with bias = 0.14 mm °C$^{-1}$ d$^{-1}$ and RMSE = 1.12 mm °C$^{-1}$ d$^{-1}$ that however only takes into account periods without new snow events, since fresh snow increases the cold content of the snowpack and contradicts the condition of the snowpack being ripe and isothermal at 0 °C. If, under the constraint of limited data availability, also changes in the cold content of the snowpack shall be considered, further research is needed on an approach that sufficiently parameterises the diurnal dynamic of vertical temperature distribution in the snowpack.

Furthermore, it is neither intended to use these *DDF* estimates directly as a model parameter nor to incorporate an energy balance based *DDF* approach into a degree-day model. One important aspect of temperature-index models is, that the *DDF* is a lumped parameter, which is usually subject to calibration

and accounts for uncertainties in different variables and parameters, e.g. temperature estimates, runoff coefficients, etc. Thus, the *DDF* estimated by the energy balance approach are rather aimed to validate the results of parameter calibration or to indicate necessary adjustments due to climate change.

The analysis of the energy balance processes controlling snowmelt indicates that cloud cover is the most decisive factor for the dynamics of the *DDF*. Under overcast conditions, the contribution of shortwave radiation is comparatively low whereas the other components are in general small. Therefore, total *DDF* is moderate and variations due to other factors are usually limited, apart from exceptional rainstorm events, for which however energy balance models are the more suitable approach.

Under clear sky conditions on the other hand, shortwave radiation is the most prominent component contributing to melt. The increase of solar angle while the melt season progresses in combination with declining albedo and a decreasing cooling effect by the longwave radiation component along with increasing air temperature leads to a pronounced temporal dynamic in the *DDF*. Whereas incoming solar radiation and net longwave radiation can be determined fairly accurate under clear sky conditions,
albedo becomes the crucial parameter for estimating the *DDF*, especially when new snow events occur during the melt period.

Clear sky conditions promote the effect of increasing *DDF* with altitude if similar melting conditions are compared, since melting temperatures arrive later in the season at higher altitudes. The opposite effect can be observed with regard to climate change. It is well known and because of higher temperature
evident, that at a certain altitude climate change will shift the snowmelt season earlier in the year. Consequently, when comparing periods of similar degree-days, as results from this study indicate the *DDFs* are expected to decrease, since solar radiation is lower and albedo is likely to be higher.

Therefore, and as pointed out by many researchers, the *DDF* cannot be considered a constant model parameter. Rather, its spatial and temporal variability must be taken into account especially when using
temperature-index models for forecasting present or predicting future water availability.

*Author contributions*

**MFI:** Conceptualization, Methodology, Software, Formal analysis, Investigation, Data curation, Writing – original draft, editing, Visualization. **WB:** Conceptualization, Methodology, Software, Formal analysis, Investigation, Data curation, Writing – reviewing & editing. **MD:** Supervision, review & editing. **MS:** Data curation, review & editing, **LK:** Supervision, Data curation.

*Competing Interest*

The contact author has declared that neither they nor their co-authors have any competing interests.

*Acknowledgements*

We would like to thank Technical University of Munich for funding the article. We also thank Koblenz University of Applied Sciences for funding the snow and meteorological station.

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
