# Peer review of "Estimating degree-day factors of snow based on energy flux components"

_The Cryosphere, 2022_

## Author Response (AR1)

Author's response accompanying revised version of 'Estimating degree-day factors based on energy flux components'. This response is set out in the following way. Firstly, updated responses to the four reviewers are provided secondly; a complete list of changes to the document is presented.

**Estimating degree-day factors based on energy flux components**
**Referee #1: Roger Braithwaite (Comments and Responses/revision)**

**Substantive Comments**

Ismail and others (submitted) is an interesting article and is very timely as many of us are concerned about the increased melting of snow and ice and its effects on streamflow. The basic premises and ambitions of the study are well presented in the ABSTRACT and in the INTRODUCTION.

The basic idea is to explain the empirical degree-day approach in terms of energy fluxes. This was done by Braithwaite (1995a) and the present paper does something similar but with a broader array of methods and models. Modern workers have access to detailed measurements from sophisticated monitoring systems and should use them where they can. Workers modelling historic data may have to use obsolete variables such as maximum and minimum temperatures and sunshine duration if these were measured with simple instruments.

Ismail and others (submitted) address both communities. Ismail and others (submitted) discuss the basic formulation of degree-day sums on lines 150-159. They are correct that several methods have been used in the past, but the common method of equating daily degree-day sum to the daily mean temperature if positive (or greater than the reference temperature if not 0 °C) is open to the criticism that there may be melt in part of the day even with daily mean temperature below zero (Arnold and McKay, 1964). Workers should calculate their degree-day sums from a sum of positive temperatures throughout the whole day if they have a modern data logger.

Braithwaite and Hughes (2022) suggest a new way of calculating degree-days if you only have maximum and minimum temperatures. This takes account of the daily temperature range, which can be quite large at lower latitudes, e.g., the Himalaya, and may cause degree-day factors to vary with latitude, as mentioned in line 146.

Ismail and others (submitted) is exceptional well-referenced, but I would like them to cite a 'senior' degree-day publication by Zinng (1951) that has stood the test of time.

Dear Reviewer,

Thank you very much for your constructive comments and suggestions to improve the manuscript. We are very grateful for the comprehensive manuscript summary and acknowledging our contribution. We shall incorporate all the necessary new references including Zinng (1951). We shall calculate the degree-day sums (if positive) based on hourly temperature data mentioned in Braithwaite and Hughes (2022).

We shall polish the language in the revised version of this manuscript. Based on your comprehensive comments and suggestion, we shall make numerous changes in the revised version of our manuscript. Below, we repeat each of your comment and our reply to them one by one. All responses are in blue font for clarity of reading.

We have added the suggested reference Zinng (1951) as well as updated degree-day sum based on hourly temperature data as suggested by Braithwaite and Hughes (2022). We have also tried to polish the language in the revised manuscript.

Muhammad Fraz Ismail

On behalf of all the authors

**USE OF ENGLISH AND RERENCING**

Ismail and others (submitted) is well written, but I wish they would use active verbs more often, and they do overuse 'however'. The text may be about 25% too long and they should remove padding and re-arrange text, so any issue is only addressed once. The reference list is accurate except for leaving out names of journals in some places, which may be an artefact of citing on-line journals.

We shall make use of more active verbs in the revised manuscript as well as minimize the use of specific words. We shall update the reference list including the journal names wherever it is missing in the list.

We make use of more active verbs in the revised manuscript as well as updated the reference list.

**DETAILED POINTS**

Line 24: define BIAS and RMSE the first time they occur.

We acknowledge that we made a typing mistake and wrote the bias in capital letters which was creating confusion. Bias is calculated by taking the average of observed – simulated. We shall clarify bias as well as define Root Mean Square Error (RMSE) in the revised manuscript.

We have now updated bias (wording) and added Root Mean Square Error instead of just writing RMSE.

Lines 25-26: Better to say 'cloud cover and snow albedo under clear sky'

We shall update it in the revised manuscript. Done

Lines 30-32: Good point!

Thank you. Done

Line 36: 'main' is better than 'unique'

We shall update it in the revised manuscript. Done

Line 41: 'more' is better than 'most'.

We shall update it in the revised manuscript. Done

Line 52: add citation to Braithwaite (1995a) here.

We shall add citation in the revised manuscript. Done

Line 88: According to Braithwaite (1995a) degree-day factors depend on mean temperatures

We shall update it in the revised manuscript.

We have re-arranged this paragraph after the comments and suggestions of reviewer 2.

Lines 105-115: Good!

Thank you. Done

Table 1. Some variables should be defined in caption or in a foot note

We shall add an explanation of each parameter in footnote for Table 1 in the revised manuscript. The footnote reads as follows.

$T_a$ = Air temperature
$P$ = Precipitation
$u$ = Wind speed
$RH$ = Relative Humidity
$A$ = Albedo (*only considered when ground is snow covered*)
$K_T$ = Clearness index
$SR_{in}$ = Incoming shortwave radiation

Done

Figure 2: Is 'Wolfgang Bogacki, 2016' reference to a publication?

It is not a reference to a publication. The picture was taken in November 2016. In the revised manuscript, we shall delete the year from image credit which causes the confusion. Done

Line 147: 'following' is better than 'along'

We shall update it in the revised manuscript. Done

Lines 151-159: I already mentioned this

Thank you very much for the comment. We shall provide the necessary reference. Done

Lines 168-169: They should not have done this! From my own thinking about the data used by Braithwaite (1995a) I am quite sure that degree-day factors are only valid for periods of many days, e.g., 10-20 days when you might expect a combination of different weather conditions and when day-to-day measurement errors may compensate.

Thank you very much for your comment and necessary clarification. Done

Line 180: much better to say 'largest' and not 'most important' as this has caused lots of problems in the literature since about 1952.

We shall update it in the revised manuscript. Done

Line 192: 'although' is better than 'however'. This occurs in a few places.

We shall update it throughout the manuscript in the revised version. Done

Line 196: 'rigorous' is better than 'rigid' and 'but' is better than 'however'

We shall update it in the revised manuscript. Done

Line 200: 'Day of the year' is a modern muddle as 1 January is day 1 in the usual counting. This means that 12:00 on 1 January is day=1.5, which is obviously wrong! Sorry!

We agree there are different definitions of 'Day of the year' by different authors which causes confusion, in particular it has not to be confused with the modern definition of Julian day. We shall clarify, that in eq. 9 (as defined by Masters, 2004), J=1 on 1st January. Done

Line 209: Should 'attenuation' be 'reflection'?

Yes, we shall update it in the revised manuscript. Done

Line 215-218. The Prescott equation is useful for historic data but not needed for modern instruments

Thank you very much for your comment and necessary clarification. We have used this, as the effect of cloud-cover can nicely be demonstrated, but we have mentioned other equations (depending on diurnal temperature variation) as well. If one has the data to apply a more sophisticated sky-model, these results can also go into the *DDF* estimates. Done

Line 226: 'when' is better than 'that'

We shall update it in the revised manuscript. Done

Lines 233-239: Very comprehensive!

Thank you. Done

Line 319: Better is 'the sensible heat component depends mainly on high wind speed and temperature' because it uses an active verb.

We shall update it in the revised manuscript. Done

Line 321: better 'is smaller on average than…'. The point is that sensible heat flux is generally smaller than the radiation components in most snowmelt situations, but sensible heat fluxes changes by a greater amount if you change temperature by 1 °C.

Thank you very much for your detailed comment and important clarification. We shall mention it in the revised manuscript as well. Done

Lines 352-353. Latent heat flux is generally a heat source to the ice/snow surface in South Greenland and a heat sink in North Greenland. This is explained by variations in vapour pressure and temperature.

Thank you very much for your detailed comment and important clarification. Done

Line 374: do you mean '… such events are rare…'?

Yes, the revised sentence shall be 'such events are rare and occur only for a brief time period'.

Revised sentence is 'such events are rare and of limited duration'.

Lines 392-394: Is this a small limitation?

Yes, this is a limitation and should be subjected to further research as mentioned in the conclusions. Done

Line 415: I was confused by the start of a new chapter here. You probably mean 'Results from Brunnenkopfhütte'. This brings me to a small concern. I accept this paper is much more than a data report from a single location, and I applaud this, but it is difficult to keep track of what material relates to which. Location. Please consider restructuring, e.g., you could discuss ALL results from Brunnenkopfhütte either before or after discussion of the more general modelling.

We shall restructure the results and discussion section in the revised manuscript. We shall mention general results as well as site specific results in different sections in order to make it clearer.

We have now restructured the manuscript.

Line 419: You should base your degree-day sum on hourly data (if positive) from your nice AWS in Fig. 2. See Braithwaite and Hughes (2022).

Based on your suggestion to use only positive degree-day sum (i.e. hourly data) from the AWS. We have now updated figure 8 for the revised manuscript. Done

[Figure]

Data source: Brunnenkopfhütte - [Period: 2016/17-2020/21]

Line 430: That confusing 'most important' again.

We shall update it in the revised manuscript. Done

**Chapters 4 and 5**:

I am confused by all the examples given. Could you not define a few 'typical' cases and give energy flux values for each case? In general, I think both chapters would benefit from some smoothing. This is something you can do more easily 1-2 months after you have written the original text.

We shall restructure the results and discussion section in the revised manuscript.

We have now restructured the manuscript.

Lines 557-561: I think this is correct, but you could phrase it better!

We shall update it in the revised manuscript. Done

Line 563: I know what RMSE means but what is BIAS? You should define all acronyms first time you use them.

We acknowledge that we made a typing mistake and wrote the bias in capital letters which was creating confusion. Bias is calculated by taking the average of observed – simulated. We shall clarify bias in the revised manuscript. Done

Figure 8: I like it. Braithwaite (1995a) should have done this for all the months in his study rather than just comparing grand-means of measured and simulate degree-day factors. I am thinking about a new paper on my old data and I will certainly make a figure like this.

We are grateful for your comment on Figure 8.

In the revised manuscript, it is now Figure 11 and moved to discussion section after comments and suggestion of several reviewers.

**Chapter 5**.

I like this. Braithwaite (1995b) looked in detail at the effect of stability on sensible heat flux model used by Braithwaite (1995a). The sensible (and latent heat) fluxes depend the density of air at the altitude in question so the degree-day factor should depend on altitude, and on latitude as lower latitude glaciers occur at greater altitude. There should be a greater latitude effect on degree-day factors than we have discovered so far. If not, why not?

We agree with the reviewer that sensible and latent heat fluxes depend on the density of air at the altitude in question. We shall consider this comment and will evaluate the effect in the respective examples/calculations.

The question concerning the influence of latitude, we considered the same melting conditions (i.e. same temperature) and same altitude which shows that at the same conditions there is only a limited influence of latitude. We shall make this clearer in the revised manuscript here and consider the effect of altitude on air density / latitude on glaciers in the discussion section.

We have added the influence of air density with changing altitude in the revised manuscript.

Lines 612-617: Interesting!

Thank you! Done

Line 630-633. Walter Ambach is the master of albedo under overcast conditions. In Braithwaite (1995a) this is one factor that reduces the time-variability of the net radiation flux.

Thank you very much for your comment. We have already cited the work done by Walter Ambach. We shall also discuss his results in the revised manuscript. Done

Line 654: this should be 'breaking in'.

We shall update it in the revised manuscript. Done

Line 665-19. I think you well explain here the importance of rain on snow.

Thank you very much for your comment. Done

Section 5.6. Ingenious!

Thank you for your encouragement. Done

Section 5.7: Although Braithwaite (1995a) clearly showed the change of degree-day factor with changing energy balances, he assumed constant degree-day factors for climate change

projections in his later papers. (I am not going to give references here as you already have too many!)

Thank you very much for the information. We shall provide the necessary references in the revised manuscript. Done

**Acknowledgements**

Was there no funding? No good advice from somebody?

We shall updated the funding source as 'Hochschule Koblenz University of Applied Sciences' and 'Technical University of Munich' in the revised manuscript. Done

**REFERENCES CITED IN THIS REVIEW**

- Arnold, KC and DK MacKay. 1964. Different methods of calculating mean daily temperatures, their effects on degree-day totals in the high Arctic and their significance to glaciology. Geographical Bulletin 21, 123-129.
- Braithwaite RJ 1995a. Positive degree-day factors for ablation on the Greenland ice sheet studied by energy-balance modelling. Journal of Glaciology 41, 137, 153-160.
- Braithwaite RJ 1995b. Aerodynamic stability and turbulent sensible-heat flux over a melting ice surface, the Greenland ice sheet. Journal of Glaciology 41, 139, 562-570.
- Braithwaite RJ and PD Hughes 2022. Positive degree-day sums in the Alps: a direct link between glacier melt and international climate policy. Journal of Glaciology1-11. http://doi.org/10.1017/jog.2021.140
- Zinng T 1951. Beziehung zwischen Temperatur und Schmelzwasser und Bedeutung für Niederschlags- und Abflüssfragen. International Association of Scientific Hydrology Publications 32, 1, 266-269.

Thank you very much for providing the list of important references we shall include/update these in the revised version of the manuscript.

References:

Masters, G. M.: Renewable and efficient electric power systems, John Wiley & Sons, Hoboken, NJ, 654 pp., 2004.

**Estimating degree-day factors based on energy flux components**
**Referee #2: Lander Van Tricht (Comments and Responses/revision)**

**General Comments**

The manuscript describes the possibility to estimate degree-day factors based on energy flux components. It studies in detail the contribution of each component as well as the variation (spatial, temporal, climate change). Consequently, the study is a valuable contribution in the context of calibrating DDF in temperature-index models to better represent melting. This is relevant given the importance of correctly calibrated models to assess (future) snowmelt.

The paper is well written, and the methods/formulations are clearly described. Further, the main ideas are very well presented in the introduction which ensures that the reader is immediately introduced in the topic and knows what the study focuses on. The study also contains an enormous number of references and (explanations of) parametrisations that sometimes make it read like a literature review, especially in the method section. The study is not particularly "innovative", but it does contribute to a better understanding of DDF and the implementation and calibration of these factors in models that can be used to determine snowmelt.

In conclusion, I think the study is worth publishing with some smaller (technical) revisions. Further, the authors may consider making the structure/division of method - results - discussion a bit clearer. Now it is not entirely clear what certain datasets are used for in this study (Brunnenkopfhütte, Upper Indus Basin, etc.). Furthermore, it could be an option to do an analysis with the hourly temperature data instead of just looking at the average, as this data is available from the meteorological station.

Dear Reviewer,

Thank you very much for your helpful comments and suggestions to improve the manuscript. We are very grateful for the manuscript summary. As suggested, we shall restructure the results and discussion section in order to make it clearer. We have estimated degree-days based on hourly data which is then aggregated to daily and then 10-daily values as mentioned in the model comparison section. We have summarized our data on daily basis because degree-day factors are estimated on daily basis.

Based on your comments and suggestions, we shall now make numerous changes in the revised version of our manuscript. Below, we repeat each of your comment and our reply to them one by one. All responses are in blue font for clarity of reading.

We have updated the suggested changes in the revised manuscript as well as restructured the results and discussion section.

Muhammad Fraz Ismail

On behalf of all the authors

**Specific comments**

Line 23: yields <-> yielded

We shall update it in the revised manuscript. Done

Line 24: mm w.e.? If water equivalent is used, use this abbreviation

Thank you for your comment. We shall updated the y-axis label in Figure 7, where it was missing. Here we have used the units as mentioned in the literature (e.g. Braithwaite and Hughes, 2021, Hock, 2003). In our opinion, the *DDF* are representing the melt so the units should be in mm instead of mm w.e.

In the revised manuscript, Figure 3 shows snow water equivalent. Done

Line 24: What is BIAS? RSME is clear for most readers. Use the full notation, especially the first time.

We acknowledge that we made a typing mistake and wrote the bias in capital letters which was creating confusion. Bias is calculated by taking the average of observed – simulated. We shall clarify it in the revised manuscript.

Updated in the revised manuscript.

Line 45: Odd use of however in this sentence

We shall replace it with modify it in the revised manuscript.

We have replaced it with 'but'.

Line 61-66: Some repetition with previous paragraphs. Consider integrating this a little more in other paragraphs. That way, the text can also become a bit shorter.

We shall integrate the text in the previous paragraphs so that there shall be no repetition.

We have integrated in the other paragraphs as suggested.

Line 88: Why does albedo decrease with increasing altitude?

Albedo is not decreasing with increasing altitude. In this sentence, it was mentioned that the degree-day factor increases with decreasing albedo. We shall clarify this sentence in the revised manuscript.

We have written an updated text to clarify our point.

Line 96: .. and topographic settings?

As suggested, we shall update it in the revised manuscript. Done

Line 117: a part of "the" Isar River system "lying" in the …

As suggested, we shall update it in the revised manuscript. The revised sentence shall read "The study area covers the Dreisäulerbach catchment, which is a part of the Isar River system and lies in the sub-alpine region of Bavaria in the Ammergauer Alps, Germany". Done

Line 122: made up sounds a bit strange. Is mainly composed or characterised?

As suggested, we shall update it in the revised manuscript. We shall use word 'characterised'. Done

Line 123: A reference here is not essential.

We shall delete the reference here in the revised manuscript. Done

Line 128: Have <-> has

We shall update it in the revised manuscript. Done

Line 130: Sometimes British – American English is used (parametrise – parametrize etc.)

Thank you for your comment. We shall update it in the revised manuscript and use only 'British English'. Done

Line 130: Summarizes

We shall update it in the revised manuscript. Done

Table 1: Some variables need explanation. What is Kt? SRin?

We shall add an explanation of each parameter in footnote in the revised manuscript. The footnote reads as follows.

$T_a$ = Air temperature
$P$ = Precipitation
$u$ = Wind speed
$RH$ = Relative Humidity
$A$ = Albedo (*only considered when ground is snow covered*)
$K_T$ = Clearness index
$SR_{in}$ = Incoming shortwave radiation

Done

Figure 1: Snow station or meteorological station?

We shall use '*Automatic snow and weather station*' in the revised manuscript. Done

Line 151: Units are in water equivalent?

The units are in '*mm*' because it refers to melt. Done

Line 155: What is the difference between part 1 and part 2 of this sentence? "T is set to 0°C" vs "The freezing point is chosen."

Thank you very much for your comment. We shall add the following sentence '$T_{DD}$ is set to 0°C'. We shall also add $T_0$ as symbol for reference temperature. Done

Line 193-194: Which value is used in this study?

We have used 1361 $W\,m^{-2}$ in this study and shall clarify this in the text. Done

Line 252: Odd use of however. Use a different word or rephrase the (part of the) sentence.

We shall update it in the revised manuscript Done

Line 294: Parametrise vs parametrize

We shall update it with British English in the revised manuscript. Done

Line 304: Parametrise vs parametrize

We shall update it with British English in the revised manuscript. Done

Line 324: It would be interesting to also mention a typical value for these conditions (W m$^{-2}$).

As suggested, we shall add typical values in the revised manuscript. Done

Line 391: Analysed vs analyzed

We shall update it with British English in the revised manuscript. Done

Line 419-420: This is based on data of the Hutt? How is the mean calculated?

Present section 4.6 of the manuscript is based on the data from Brunnenkopfhütte station. The data is available on 10-minutes interval which is then aggregated on mean hourly and daily basis. We shall clarify different data sources and their respective use in the revised manuscript. Done

Line 470: I think it is clearer to put the panel letter before the sentence.

As suggested, we shall update it in the revised manuscript. Done

Line 474: Snow station or meteorological station?

We shall use '*Automatic snow and weather station*' throughout the manuscript. Done

Line 488-489: An average temperature of 20°C, it is not very common in early spring, right?

We agree that in early spring an average temperature of 20 °C is not common. But here we have used a broad range of degree-days for our illustrative examples and summary tables. Done

Line 492 and Figure 5: for selected cloudiness and average air temperatures?

Thank you for your suggestion. We shall change it in Figure 5 as 'for selected cloudiness [%] and degree-days [°C d]. Done

Line 510-512: Would it be an option to derive an average using the average hourly wind speeds?

The automatic snow and weather station data has temporal resolution of 10-minutes. This data is then aggregated on hourly and then daily basis for analysis purpose. Done

Line 539: I prefer "refreezes" <-> is refrozen

As suggested, we shall update it with 'refreezes' in the revised manuscript. Done

Line 554: meteorological station <-> snow station

We shall use 'Automatic snow and weather station' throughout the manuscript. Done

Line 563: What is BIAS? Use full notation the first time

We made a typing mistake and wrote the bias in capital letters which was creating confusion. Bias is calculated by taking the average of observed – simulated. We shall make it clear in the revised manuscript. Done

Line 563-565: The snowmelt periods which are neglected, are these particular days? Or 10-day periods?

Those 10-daily snowmelt periods in which a new snow event occurred (marked by hollow circles in Figure 8) were excluded from the calculation of the error metrics.

In the revised manuscript, Figure 8 is now Figure 11 and moved to discussion section after comments and suggestions of other reviewers.

Line 582: "is" or "to be"

We shall update it with 'to be' in the revised manuscript. Done

Line 704-706: Where does this data come from? The area of Indus Basin is not elsewhere introduced or mentioned.)

We shall update the dataset as well as Upper Indus Basin (UIB) catchment information in the revised version of this manuscript. For climate change impact assessment (i.e. temperature) the bias-corrected climate scenarios from four GCMs (GFDL-ESM2M, HadGEM2-ES, IPSL-CM5A-LR, MIROC5) driven by two representative concentration pathways (RCPs), which were provided by the ISIMIP project (Hempel et al. 2013; Frieler et al. 2017) were used (Ismail et al. 2020).

- Frieler, K., Lange, S., Piontek, F., Reyer, C. P. O., Schewe, J., Warszawski, L., Zhao, F., Chini, L., Denvil, S., Emanuel, K., Geiger, T., Halladay, K., Hurtt, G., Mengel, M., Murakami, D., Ostberg, S., Popp, A., Riva, R., Stevanovic, M., Suzuki, T., Volkholz, J., Burke, E., Ciais, P., Ebi, K., Eddy, T. D., Elliott, J., Galbraith, E., Gosling, S. N., Hattermann, F., Hickler, T., Hinkel, J., Hof, C., Huber, V., Jägermeyr, J., Krysanova, V., Marcé, R., Müller Schmied, H., Mouratiadou, I., Pierson, D., Tittensor, D. P., Vautard, R., van Vliet, M., Biber, M. F., Betts, R. A., Bodirsky, B. L., Deryng, D., Frolking, S., Jones, C. D., Lotze, H. K., Lotze-Campen, H., Sahajpal, R., Thonicke, K., Tian, H., and Yamagata, Y.: Assessing the impacts of 1.5 °C global warming – simulation protocol of the Inter-Sectoral Impact Model Intercomparison Project (ISIMIP2b), Geosci. Model Dev., 10, 4321–4345, https://doi.org/10.5194/gmd-10-4321-2017, 2017.
- Hempel, S., Frieler, K., Warszawski, L., Schewe, J., and Piontek, F.: A trend-preserving bias correction – the ISI-MIP approach, Earth Syst. Dynam., 4, 219–236, https://doi.org/10.5194/esd-4-219-2013, 2013.
- Ismail, M.F., Naz, B.S., Wortmann, M. et al. Comparison of two model calibration approaches and their influence on future projections under climate change in the Upper Indus Basin. Climatic Change 163, 1227–1246 (2020). https://doi.org/10.1007/s10584-020-02902-3

We have added a new data section 2.2 in the revised manuscript.

Line 734: Parametrizes <-> parametrises (probably I have missed other ones)

We shall update it with British English in the revised manuscript. Done

**Estimating degree-day factors based on energy flux components**
**Referee #3: Rijan Kayastha (Comments and Responses/revision)**

**General Comments**

This paper tries to do something new on the positive degree-day factor by analysing different previous research which is very good. It is good that the authors still agree that the conventional degree-day approach is still good to use where data are insufficient. I have found the paper deals with the shortwave radiation calculation in detail which is very good for data insufficient regions. But the others such as the need of using different degree-day factor for space and time has already been applied in many previous researches and need to mention in this study. I also like to comment on the symbol used for a degree-day factor; in the past papers degree-day factor is denoted by the letter "k or K" but nowadays DDF is being used. The authors should also think about this issue. About the use of the degree-day factor in a climate change study, if we consider all parameters which affect the degree-day factor and assign the degree-day factor accordingly, it will still give a good result. Authors should also think about it.

Dear Reviewer,

Thank you very much for your helpful comments and suggestions to improve the manuscript. We are very grateful for the manuscript summary. In the revised manuscript, we shall mention the related studies where the authors highlighted the need of using degree-day factors for space and time. We agree that in the past for denoting the degree-day factor symbol '$k$' (e.g. Braithwaite, (1995b) or '$a$' (Rango and Martinec, 1995)) has been used. But in this study we used '$DDF$' because '$k$' has been already mentioned for von Karman's constant. We agree that in climate change studies if we consider all the parameters affecting the degree-day factor, it will give good results. But in present study we have tried to highlight that how the degree-day factors might vary under climate change, keeping in view the data constraints. In our opinion, if comprehensive dataset is available then it would be appropriate to use energy balance models. Of course, it makes sense to estimate the influence of each effecting parameter on the $DDF$.

Keeping in view all of your comments and suggestion, we shall make numerous changes in the revised version of our manuscript. Below, we repeat each of your comment and our reply to them one by one. All responses are in blue font for clarity of reading.

All the suggested changed has been incorporated in the revised manuscript.

Muhammad Fraz Ismail
On behalf of all the authors

**Specific comments**

Line 118: Need to mention the name of the country (Germany) after Ammergauer Alps.

We shall add the country name in the revised manuscript. Done

Line 260: It should be "The net longwave radiation flux …….

We shall update it in the revised manuscript. Done

Line 261: Equation (20) should be at line 264 instead of line 261 at present. The sentence does not look good at present.

In the revised manuscript we shall place equation at line 264. Done

Line 233: Need to use a different letter for a coefficient other than k. Because k is used as Von Karmann constant on line 310.

We agree and it will be replaced with '$k_{Rs}$' in the revised manuscript.

We have used $k_H$ in the revised manuscript.

Line 404: should be degree-day models instead of "degree-day factor models."

Thank you for your comment. We shall update it in the revised manuscript. Done

Line 461-463: The result stated in those lines "All of these models show the same tendency of linear increase by altitude, with the altitude factor being comparatively smaller under clear sky compared to overcast conditions" is to some extent is different from the results which we have received on a Glacier AX010 in Nepal (Kayastha et al., 2000).

Actually, figure 4 is showing the relative increase in the altitude factor depending upon sky conditions (i.e. for clear sky $K_T = 0.75$ and under overcast condition $K_T = 0.25$). For the same elevation difference, the absolute change in clear sky is greater compared to overcast condition. We agree that we shall clarify this point in the revised manuscript.

Following text has been added in the revised manuscript.

"It should be noted, that although $K_z$ is higher for overcast than for clear sky conditions, the absolute increase of the clearness index $K_T$ with altitude is higher under clear sky conditions."

Figure 10 shows that the degree-day factor at higher altitudes is higher in a comparative clear sky (in June) compared to July and August (peak monsoon season with a highly overcast period in Nepal). We assumed that due to the overcast situation, air temp does not change much and hence degree-day factors too do not change much. Why in the present study is the altitude factor comparatively smaller under the clear sky?

In Figure 10, we have tried to show the expected changes in the degree-day factors based on projected climate change (i.e. in this case temperature change). In this particular case, we have kept sky conditions as constant (i.e. clear sky). In addition, we have not applied any altitude factor in this specific case like we have done in figure 9 (b). But if we apply the clearness altitude factor then it would change the results as shown in the following figure. If sky conditions are changed then of course it will also alter the degree-day factor.

We agree that July and August are the peak monsoon season in this region with a highly overcast periods, so it will definitely impact the degree-day factors. We think that your comment here about the altitude factor is related to figure 4 which has been answered in the previous question. Done

[Figure]

Line 639 -640: This statement "Under overcast conditions, however, the DDF is virtually stable ranging from 4.4 to 4.5 mm °C-1 d-1 in the same period" is in agreement with what was shown in Figure 10 in Kayastha et al. (2000).

Thank you for your comment. We shall add the necessary citation in the revised manuscript. Reference added in the revised manuscript.

Line 760-761: The message of this statement "Therefore, and as pointed out by many researchers, the DDF cannot be considered a constant model parameter. Rather, its spatial and temporal variability must be taken into account …." Has already been implemented in Kayastha et al. (2020; Table 3) in which we have used two sets of degree-day factors; lower degree-day factor at lower altitudes (lower than 5000 m) and higher degree-day factor for higher altitudes (above 5000 m). Also, monthly degree-day factors are used to incorporate the seasonality of degree-day factors.

Thank you for your comment and necessary clarification. We shall add the important citation in the revised manuscript. We agree that it is important to consider the spatial and temporal variation in the degree-day factors.

In the revised manuscript, we have added the required references in the discussion section.

References:

Kayastha, R. B., Ageta, Y. & Nakawo, M. (2000). Positive degree-day factors for ablation on glaciers in the Nepalese Himalayas: case study on Glacier AX010 in Shorong Himal, Nepal. Bulletin of Glaciological Research, 17, 1-10.

Kayastha, R. B. & Kayastha, R. (2020). Glacio-Hydrological Degree-Day Model (GDM) Useful for the Himalayan River Basins. In: Dimri A., Bookhagen B., Stoffel M., Yasunari T. (eds) Himalayan Weather and Climate and their Impact on the Environment. Springer, Cham, Doi: 10.1007/978-3-030-29684-1_19.

We shall add the important references in the introduction section as well. Done

References:

- Rango, Albert and Jaroslav Martinec. "Revisiting the Degree-Day Method for Snowmelt Computations." Journal of The American Water Resources Association 31 (1995): 657-669.
- Braithwaite, R. J. 1995b. Positive degree-day factors for ablation on the Greenland ice sheet studied by energy-balance modelling. J. Glaciol, 41(137), 153–160.

**Estimating degree-day factors based on energy flux components**
**Referee #4: Álvaro Ayala (Comments and Responses/revision)**

**PAPER SUMMARY AND RECOMMENDATION**

Ismail and co-authors investigate how degree-day factors (DDFs) depend on the components of the snowpack energy balance. Assuming a snowpack close to melting conditions and a negligible cold content, the authors connect DDFs to the variations of each energy balance component by means of a set of widely used equations. In this way, DDFs are related with different characteristics and conditions, such as elevation, latitude and meteorological variables. The authors provide several summary tables and figures that can be used by other researchers to estimate DDFs in poorly monitored regions using minimum data requirements. Additionally, the authors estimate the impact of climate change on DDFs. They conclude that cloud cover and snow albedo are the main processes controlling DDFs and that DDFs cannot be treated as constant parameters.

The study is appropriate for The Cryosphere. The article is well written, but some parts describing the equations can be shortened. I think that the authors do a valuable contribution. Having tools to estimate DDFs is a good idea, and it can be useful for researchers working on the snow hydrology of poorly monitored regions. However, I think that the article needs to be improved before being suitable for publication. Please see my main comments.

Dear Reviewer,

Thank you very much for your comments and suggestions to improve the manuscript. As suggested, we shall restructure the results and discussion section in order to make it clearer. We shall make numerous changes in the revised version of our manuscript. Below, we repeat each of your comment and our reply to them one by one. All responses are in blue font for clarity of reading.

We have now restructured the revised manuscript and incorporated the suggested changes.

Muhammad Fraz Ismail

On behalf of all the authors

**MAJOR COMMENTS**

**1. Presentation and role of the datasets**

Field dataset: The purpose of including the datasets from Brunnenkopfhütte and Naran stations is not clearly presented. The authors should mention in the Introduction what is the role of these datasets in their study. Are they used as validation, or test sites? Do the authors make tests at the catchment or point scales? Importantly, the use of the Naran dataset comes a surprise in the middle of the discussion section.

Climate change dataset: Please provide more details about this dataset and add this analysis to the objectives of the study.

The main purpose of using the Brunnenkopfhütte snow station data is to show how the degree-day factor can be estimated under naturally varying hydro-meteorological conditions in the

field. We shall clearly mention the purpose of these datasets in the data section. The Brunnenkopfhütte station is our test site where we have installed our snow and meteorological station. We have done a point scale analysis based on the datasets from Brunnenkopfhütte test site.

But when we are discussing the Naran station as well as Upper Indus area then our aim is to address the problem related to estimate DDFs in poorly monitored regions, where only limited data is available. We shall clarify these points in the revised manuscript.

For climate change impact assessment (i.e. temperature) the bias-corrected climate scenarios from four GCMs (GFDL-ESM2M, HadGEM2-ES, IPSL-CM5A-LR, MIROC5) driven by two representative concentration pathways (RCPs), which were provided by the ISIMIP project (Hempel et al. 2013; Frieler et al. 2017) were used (Ismail et al. 2020). We shall also clarify this point in the revised manuscript.

- Frieler, K., Lange, S., Piontek, F., Reyer, C. P. O., Schewe, J., Warszawski, L., Zhao, F., Chini, L., Denvil, S., Emanuel, K., Geiger, T., Halladay, K., Hurtt, G., Mengel, M., Murakami, D., Ostberg, S., Popp, A., Riva, R., Stevanovic, M., Suzuki, T., Volkholz, J., Burke, E., Ciais, P., Ebi, K., Eddy, T. D., Elliott, J., Galbraith, E., Gosling, S. N., Hattermann, F., Hickler, T., Hinkel, J., Hof, C., Huber, V., Jägermeyr, J., Krysanova, V., Marcé, R., Müller Schmied, H., Mouratiadou, I., Pierson, D., Tittensor, D. P., Vautard, R., van Vliet, M., Biber, M. F., Betts, R. A., Bodirsky, B. L., Deryng, D., Frolking, S., Jones, C. D., Lotze, H. K., Lotze-Campen, H., Sahajpal, R., Thonicke, K., Tian, H., and Yamagata, Y.: Assessing the impacts of 1.5 °C global warming – simulation protocol of the Inter-Sectoral Impact Model Intercomparison Project (ISIMIP2b), Geosci. Model Dev., 10, 4321–4345, https://doi.org/10.5194/gmd-10-4321-2017, 2017.
- Hempel, S., Frieler, K., Warszawski, L., Schewe, J., and Piontek, F.: A trend-preserving bias correction – the ISI-MIP approach, Earth Syst. Dynam., 4, 219–236, https://doi.org/10.5194/esd-4-219-2013, 2013.
- Ismail, M.F., Naz, B.S., Wortmann, M. et al. Comparison of two model calibration approaches and their influence on future projections under climate change in the Upper Indus Basin. Climatic Change 163, 1227–1246 (2020). https://doi.org/10.1007/s10584-020-02902-3

In view of your comments and suggestions, we have now bifurcated the section 2 into two parts. (i) Test site (ii) Datasets. We have now clearly stated the aim of using different datasets.

**2.2 Datasets**

Present study utilises three different dataset. Data sources as well as aim of using these datasets are mentioned as follow:

- We use observed hydro-meteorological datasets from a test site (i.e. Brunnenkopfhütte) with the aim to show how the DDF can be estimated for a specific site under naturally varying hydro-meteorological conditions.
- In order to demonstrate the variation of the DDF over time, location, and altitude as well as its significance for temperature-index modelling, we use elevation zone-wise temperature data of the Upper Jhelum Basin from a previous study (Bogacki and Ismail, 2016).

- In the discussion section (Sec. 5), we perform a brief analysis in order to show the influence of climate change on the DDF in poorly monitored regions, for example Himalayas-Karakoram-Hindukush (i.e. Upper Indus Basin). In this specific analysis, projected changes in temperature are based on a previous study (Ismail et al., 2020). These projected changes in temperature are the median of four GCMs (GFDL-ESM2M, HadGEM2-ES, IPSL-CM5A-LR, and MIROC5) that are driven by two representative concentration pathways (RCP2.6 and RCP8.5). This data is provided by the Inter-Sectoral Impact Model Intercomparison Project (ISIMIP) (Hempel et al., 2013; Frieler et al., 2017).

**2. Discussion section**

In this section, the authors continue their analysis and calculations, but they provide almost no comparisons with the results of other studies. The authors should discuss their results using the literature presented in the Introduction. Additionally, I recommend the inclusion of some other references regarding the spatial and temporal transferability of degree-day factors (or temperature factors) and melt parameters that, in my opinion, are missing (Ohmura, 2001; Carenzo et al., 2009; MacDougall and Flowers, 2011; MacDougall et al., 2011; Gabbi et al., 2014). The limitations of the approach proposed by the authors and the assumptions made through the article should be more discussed. For example, the authors validate their approach using only one monitoring station, can the authors include more data? There are certainly more datasets available for which DDFs have been derived. Otherwise, this is an important limitation of the study that should be discussed.

We shall restructure the discussion section and provide more comparison insight as suggested by the reviewer. We shall also include important references specifically regarding the spatial and temporal transferability of degree-day factors.

We agree that the presented approach has its limitations, to our opinion mainly the assumption that the snowpack is isothermal at 0°C and in fully ripe state. However, the aim of the paper is not to present a new and comprehensive degree-day factor approach, which certainly would have to be validated by a number of datasets. We rather want to demonstrate how well established energy balance formulas can be applied in data scarce situations to estimate melt and translate this into degree-day factors. For this purpose, we present tools like the set of existing formulas, summary tables, and graphs and we give a number of examples in order to demonstrate the influencing factors under several spatial and meteorological conditions.

In contrast to exemplifying the individual factors, the Brunnenkopf station example shall demonstrate how these tools can be applied in a complex real-live situation and give an idea about accuracy of estimated degree-day factors. Moreover, in Figure 8 we specifically used this example to show the effect of a fully ripe snowpack vs one with a considerable cold content. Taking the limited accuracy of field derived degree-day factors, we feel it would be unsuitable to make a similar comparison with a foreign dataset ourselves without knowing all subtles of the data. However, we would be more than happy if our paper would motivate other researchers to test the presented tools with their own familiar datasets.

We have now restructured the discussion section as well as added related comparison and references to clarify our point of view.

**3. Conclusions and recommendations**

As the aim of the study is to "quantify the effects of spatial, temporal, and climatic conditions on the DDFs" and the conclusion is that "DDF cannot be treated as a constant parameter", what are the recommendations of the authors to a researcher modelling the snow hydrology of poorly monitored catchments? Should that researcher use a range of parameters from your equations? How large should be the variability of DDFs in space and time? Different DDFs for each sub-catchment, slope or elevation band? How often should the DDFs change in time? Every week, month or season? I think that the article would benefit from such discussions and recommendations.

Yes, the aim is to quantify the effects of spatial, temporal, and climatic conditions on the degree-day factors. As explained by several authors (Braithwaite 1995a, Hock, 2003, Kayastha et al. 2000), we have also recommended that the DDF cannot be considered a constant model parameter. Rather, its spatial and temporal variability must be taken into account especially when using temperature-index models for forecasting present or predicting future water availability. In section 5.6 and 5.7 of the manuscript we have tried to show that how one can estimate the degree-day factors based on only temperature data and assumed typical climatic conditions. We have presented summary tables and figures in order to get an initial idea about the range of degree-day factors based on available information.

We have showed that how the degree-day factors could change depending upon elevation. Of course if sub-catchments have different hydro-climatic conditions then it will ultimately impact the degree-day factor. There are several recommendations on changing the degree-day factors in space and time, for example on monthly as well as on seasonal basis (Kayastha et al. 2000, Braithwaite 1995a). We presented 10-daily DDF for forecasting water availability in an operational model. We shall make it clearer in the revised manuscript.

We have now tried to make it clearer in the revised manuscript. Done

**MINOR COMMENTS FOR THE AUTHORS**

12-13: I would add "At mid-latitudes, seasonal snow …" because this seasonal pattern is not necessarily found on every snow and ice dominated mountain catchment (e.g. tropical glaciers).

Thank you very much for your comment and necessary clarification. We shall update it in the revised manuscript. Done

13: I think that the concept of snowmelt runoff is wider than what the authors are describing. The authors are describing only the process of melt whereas snowmelt runoff include other processes controlling the movement of excess meltwater through a catchment.

We agree with the reviewer on this point. We shall clarify it in the updated manuscript. We shall also remove the word 'runoff' as indicated. Done

21: is physically based -> is based

We shall update it in the revised manuscript. Done

22: I don't think that the formulas are "approximate", they just have limitations and assumptions.

We agree with the reviewer on this point. We used formulas related to minimal data requirements. We shall update it in the revised manuscript. Done

23: observed -> field-derived. DDFs cannot be measured in the field because they are not a physical quantity.

We shall update it in the revised manuscript. Done

30: "albedo is likely to be higher", there are also other reasons, such as lower radiation and temperatures, aren't they?

We are comparing period of similar degree-days so temperature will not be higher. Yes, radiation will be lower as you pointed and we shall include it in the revised manuscript. Done

35: It would be interesting to mention somewhere in the Introduction that researchers usually select DDFs values from other studies and that the spatial transferability is not always good [e.g. Carenzo et al., 2009; Wheler, 2009].

We shall update it in the revised manuscript. Done

35: The authors should briefly mention at the end of the Introduction what is the role of the Study area in the article as Section 2 "Study area" comes as a surprise. See my main comment.

We agree with the reviewer on this point. We shall clarify the role of study area as well as the data sets used. We shall update it in the revised manuscript.

New section added. Done

79: "longer time periods" Can the authors be more precise? Weeks, months, years?

We shall update it in the revised manuscript. The longer time period (i.e. 10-daily, monthly, seasonal) as mentioned by different authors (Ismail and Bogacki, 2018; Braithwaite 1995a; Kayastha et al. 2000). Done

81-82: Also, the spatial variability of air temperature does not fully describe the spatial variability of the energy balance.

Thank you very much for your comment. We shall consider this in the revised manuscript.

We have re-arranged it in the introduction.

118: Please mention the country

We shall update it in the revised manuscript. Done

123: The Kopp reference is not necessary here as the authors also have a DEM of the catchment.

We shall delete the unnecessary reference here in the revised manuscript. Done

171-172: "The balance of the energy fluxes over the surface of the snowpack". Please note that Q_G (ground heat) is not a surface flux. By including DeltaQ and Q_G, the authors are describing the energy balance of the entire snowpack and not only the surface, which has not heat capacity [den Broeke et al., 2011]. Otherwise please clearly define what control volume is considered by the authors.

Thank you very much for your comment. We shall clarify by specifying the control volume in the revised manuscript. Done

179: The length of this section can be reduced.

Thank you for your comment. We consider short wave radiation is a very important component. We shall see where it can be shortened.

We have updated this section.

182: No reference is needed for equation 5

Ok. We shall delete it. Done

241: I'm a bit confused, when the authors correct by elevation, what is the term that goes in eq. 6, K_z or K_T?

$K_T$ goes into eq. (6). $K_T = K_z \times K_{T0}$. We used the formulation in eq. 18 to provide the definition of clearness altitude factor $K_z$. Nevertheless, we shall clearly explain the clearness altitude factor in combination with Figure 4.

We have now tried to explain it clearly in the revised manuscript.

277: Please clarify at what height above the surface are Pv and Ta measured.

Thank you for your comment. We shall add that Brutsaert developed eq. 22 for $p_v$ and $T_a$ measured at screen level. In our examples, both parameters are measure at 2m above the surface. Done

300: What do the authors mean by "a probabilistic reasoning"?

We wanted to say that Badescu and Paulescu (2011) used probability distributions to develop relations between cloudiness and relative sunshine hours and showed that a linear relation is a first good estimate. We shall formulate in the revised manuscript accordingly.

The formulation has been updated in the revised manuscript.

Nevertheless, in simple sky models usually a linear relation between cloudiness and relative sunshine hours is applied as a first approximation (e.g. Brutsaert, 1982; Annandale et al., 2002; Pelkowski, 2009) which, as Badescu and Paulescu, (2011) showed by using probability distributions to develop relations 360 between cloudiness and relative sunshine hours, is a first good estimate.

344: I think a step or equation is missing here and it should be that relating RH and p0. Or how do the authors calculate pv? Also, are the authors assuming saturated conditions at the snow surface?

Thank you very much for your comment. We thought it is obvious that $p_v$ can be calculated from RH and $p_0$ but we shall include this step in the revised manuscript and shall also add that we assume saturated conditions at the snow surface.

New equation (32) added in the revised manuscript.

Knowing the relative humidity $\psi$ (-) and the saturation vapour pressure $p_s$ at a given air temperature, the actual vapour pressure $p_v$ (Pa) can be calculated through the relation

$$p_v = \psi \, p_s$$  (1)

354/375: Sections 3.2.5 and 3.2.7 don't read as "Methods". They seem a review on the subject. As both terms (Q_G and DeltaQ) are neglected by the authors, I suggest the shortening of

these sections and to move them to the beginning of Section 3.2 where a suitable justification to neglect them can be provided.

Thank you very much for your comment. We shall see where these sections can be shortened. We prefer to keep these sections here because these are in line with the equations mentioned earlier.

We have shortened the change in internal storage section.

422: Delete "approximate".

We shall delete it in the revised manuscript. Done

431: higher altitudes, as well as dry climates.

Thank you for your comment. We shall update it in the revised manuscript. Done

504: As wind speed is highly variable in space and time, I don't think that the authors can refer to "typical values". It would be better to write something such as: "… can be roughly estimated based on the topographic and climate characteristics of the study site".

Thank you for your comment. We agree and it shall be updated it in the revised manuscript. We have updated this section based on comments and suggestions of other reviewers.

551: I think that this is the first time that the authors mention the goal of these dataset. Please see my main comments.

We shall update this section in the revised manuscript as suggested in main comments.

A new section '2.2 Datasets' has been added in the revised manuscript.

579: I believe that this is not clearly a discussion section because there are almost no comparisons against other studies (and almost no references). Instead, the authors present more results and analysis. Please my main comments.

We shall update this section in the revised manuscript as suggested in main comments.

We have restructured the discussion section as well as added necessary comparison against other studies.

592: This is the first time that the authors mention these data. Please properly introduce this site and the dataset in section 2. Also explain what is the purpose of including this dataset.

We agree with the reviewer. We shall update this section in the revised manuscript as suggested in main comments.

A new section '2.2 Datasets' has been added in the revised manuscript. We have also added the purpose of using each dataset.

598: Please change the word "altitude" by "elevation" throughout the article. Altitude is the vertical distance between an object and the earth's surface.

Sometimes elevation and altitude are used interchangeably. We shall see where this wording can be used and update this in the revised manuscript. Done

606: Why does the solar angle change with altitude?

Because solar angles changes from February to May. We shall add this in the revised manuscript. Done

693-695: Not clear, please reword.

We shall make it clear in the revised manuscript. Done

702-705: This belongs to methods. The climate change analysis should be introduced earlier in the manuscript. Provide more details about these data, are those values an average of different GCMs?

We agree with the reviewer. We shall update this section in the revised manuscript as suggested in main comments.

The projected changes in temperature are median values as mention in Ismail et al. (2020). These are based on four models four GCMs (GFDL-ESM2M, HadGEM2-ES, IPSL-CM5A-LR, MIROC5) driven by two representative concentration pathways (RCPs), which were provided by the ISIMIP project (Hempel et al. 2013; Frieler et al. 2017).

A new section '2.2 Datasets' has been added in the revised manuscript. We have also added the purpose of using each dataset.

697: Musselman et al. [2017] is an excellent article regarding slower melt rates in climate change scenarios.

Thank you very much for sharing this reference. We shall cite this article in the revised version of manuscript.

We have added the necessary reference in the revised manuscript.

**SUGGESTED TECHNICAL CORRECTIONS FOR THE AUTHORS**

11: Meltwater

We shall update it in the revised manuscript. Done

11: Consider: "Meltwater from mountainous catchments dominated by snow and ice is a…"

We shall update it in the revised manuscript. Done

36: Meltwater

We shall update it in the revised manuscript. Done

42: Delete "for the prediction"

We shall update it in the revised manuscript. Done

44: Delete "runoff". The authors discuss only the process of melt.

We shall update it in the revised manuscript. Done

59: Add "using runoff" after DDF

We shall update it in the revised manuscript. Done

61: Delete "runoff"

We shall update it in the revised manuscript. Done

68: by the inclusion

We shall update it in the revised manuscript. Done

72: the position

We shall update it in the revised manuscript. Done

95: Since melt depends …

We shall update it in the revised manuscript. Done

117: system and lies

We shall update it in the revised manuscript. Done

119: delete about or ~

We shall update it in the revised manuscript. Done

127: a standard

We shall update it in the revised manuscript. Done

128: Brunnenkopfhütte site

We shall update it in the revised manuscript. Done

146: Delete "concrete", or maybe use "actual".

We shall update it in the revised manuscript. Done

221: Delete "," after disadvantage

We shall update it in the revised manuscript. Done

283: the above relation

We shall update it in the revised manuscript. Done

284: … snowpack amounts to

We shall update it in the revised manuscript. Done

294: Add "," after parameterize

We shall update it in the revised manuscript. Done

294-295: and their effects on radiation depend…

We shall update it in the revised manuscript. Done

329: the snow and the snow surface

We shall update it in the revised manuscript. Done

371: even during extreme weather conditions

We shall update it in the revised manuscript. Done

588-590: Please rewrite these lines for clarity.

We shall rewrite it in the revised manuscript. Done

591: "The example", what example?

We shall update the sentence like, 'In Figure 9 (a), we compare'. We shall add it in the revised manuscript. Done

638: see Table

We shall update it in the revised manuscript. Done

650: in Table

We shall update it in the revised manuscript. Done

**FIGURES**

Figure 1: I think that m (instead of cm) are enough for "High" and "Low" in the legend.

We shall update it in the revised manuscript as shown below. Done

[Figure]

Figure 4: Why is the clearness index (K_T) at a given elevation larger for overcast conditions than for clear sky? Shouldn't be the opposite? Please clarify.

Actually, the Figure 4 is showing the relative increase in the altitude factor depending upon sky conditions (i.e. clear sky $K_T = 0.75$ and overcast condition $K_T = 0.25$). For the same elevation difference, the absolute change in clear sky is greater compared to overcast condition. We agree that we shall clarify this point in the revised manuscript.

We have added the updated text "It should be noted, that although $K_z$ is higher for overcast than for clear sky conditions, the absolute increase of the clearness index $K_T$ with altitude is higher under clear sky conditions." Done

Figure 10: Why exactly do DDFs on each panel (present and scenarios) decrease as the season progresses but in Figure 9 DDFs increase as the season progresses?

We have already mentioned in section 5.6 of the paper that in contrast to Figure (d), the *DDF* decreases continuously in all elevation zones (i.e. Figure 10) in the subsequent melting periods. It is because air temperature and thus degree-days rise faster compared to the increase in melt. We shall make it clearer in the revised manuscript. Done

**TABLES**

Table 1: Explain the name of the variables.

We shall add an explanation of each parameter in footnote in the revised manuscript. The footnote reads as follows.

$T_a$ = Air temperature
$P$ = Precipitation
$u$ = Wind speed
$RH$ = Relative Humidity
$A$ = Albedo (*only considered when ground is snow covered*)
$K_T$ = Clearness index
$SR_{in}$ = Incoming shortwave radiation

Done

Table 1: Please provide $SR_{in}$ in W/m$^2$

We shall provide $SR_{in}$ in W m$^{-2}$ it in the revised manuscript. Done

References:

Ismail, M. F. and Bogacki, W.: Scenario approach for the seasonal forecast of Kharif flows from the Upper Indus Basin, Hydrol. Earth Syst. Sci., 22, 1391–1409, https://doi.org/10.5194/hess-22-1391-2018, 2018.

Kayastha, R. B., Ageta, Y. & Nakawo, M. (2000). Positive degree-day factors for ablation on glaciers in the Nepalese Himalayas: case study on Glacier AX010 in Shorong Himal, Nepal. Bulletin of Glaciological Research, 17, 1-10.

Braithwaite RJ 1995a. Positive degree-day factors for ablation on the Greenland ice sheet studied by energy-balance modelling. Journal of Glaciology 41, 137, 153-160.

Hock, R.: Temperature index melt modelling in mountain areas, Journal of Hydrology, 282, 104–115, https://doi.org/10.1016/S0022-1694 (03)00257-9, 2003.

**Complete list of changes to the manuscript**

In the list below, all line numbers refer to the tracked changes version of the manuscript. Changes, which relate solely to renumbering of figures, equations or sections following insertion of new elements, are not included in this list.

**Line 8**: Additional correspondence email address added.

**Line 12**: 'At mid-latitudes, seasonal' (R2)

**Line 24:** The word 'bias' is now updated before that it was BIAS which was causing confusion to all the reviewers.

**Line 31**: 'when solar radiation is lower' added (R4)

**Line 37**: 'Meltwater' added (R4)

**Line 49 – 51**: literature review updated and repetitive text deleted (R2)

**Line 52**: Reference to Zingg added (R1)

**Line 64 – 65**: Literature review updated (R4)

**Line 106**: approximate is replaced with estimated (R4)

**Line 117**: Test site and data section added according to Reviewer 4 suggestion.

**Line 120**: Germany added (R3)

**Line 125**: The area is mostly characterized (R2)

**Line 126**: Reference deleted (R2)

**Line 140**: Table 1, $SR_{in}$ values are updated with W m$^{-2}$.

**Line 145**: Footer added for variables explanation as suggested by Reviewer 1, 2 and 3.

**Line 182**: Sentence updated according to Reviewer 2 suggestion.

**Line 211**: 'the largest source' (R1)

**Line 226 – 227**: Measurements by Kopp and Lean, (2011) indicate a present value 225 of about 1361 W m$^{-2}$ (R2)

**Line 233**: 'the number, with J = 1 on January 1st.' (R1)

**Line 275**: Symbol updated after Reviewer 3 suggestion.

**Line 290 – 300**: Text updated for clarification as suggested by Reviewer 4.

**Line 372 – 381**: added new equations and text as asked by Reviewer 1 and 4.

**Line 395 – 400**: New text added after Reviewer 2 suggestion.

**Line 421 – 423**: added new equations after the comments of Reviewer 4.

**Line 455**: 'rare and of limited duration' updated as suggestion by Reviewer 1.

**Line 456**: Section 3.2.7 has been shortened as suggested by Reviewer 2 and 4.

**Line 529**: Figure 5 caption updated

**Line 551 – 572**: Section 4.3 has been updated after the comments and suggestion of Reviewer 1.

**Line 596 – 616**: Section 4.4 has been updated after the comments and suggestion of Reviewer 1.

**Line 644**: The discussion section has been updated and restructured as suggested by Reviewer 1 and 4.

**Line 692 – 699**: Added new text after the comments of Reviewer 4.

**Line 711 – 713:** Comparison added as after the comments of Reviewer 4.

**Line 725 – 728**: Added new text after the comments of Reviewer 3.

**Line 742**: 'breaking in' (R1)

**Line 771 – 772**: Analysis updated as suggested by Reviewer 1.

**Line 828 – 833:** Comparison added and text updated after the comments of Reviewer 4.

**Line 853 – 855:** Necessary reference added (R4).

**Line 913 – 915**: Funding source has been updated (R1).

---

## Editor Decision (ED1)

*[handwritten margin note, top:]* Not always clear what you use and what is given as 'extra' info. → (some) TABLES TO SUPPL. MAT + MERGE.

*[handwritten note:]* + more compact materials + methods } focus on what matters!

*[handwritten note:]* + update references!

*[handwritten red note:]* In general, need to specify this. Also in some locations in text.
Otherwise could also be ice (+for outsiders is not given that they know this typically applies to snow and ice)

[revised manuscript text omitted]

*[handwritten margin note: Quite a lot on this recently. Would be good to go a bit more in detail into this. See also general comment and work by Matthews et al. (under review), Bolibar et al. (2022), and Vincent and Thibert (2022)]*

100 water availability under climate change scenarios is typically modelled with *DDFs* calibrated for the present climate, which increases the parametric uncertainty introduced by the hydrological models (Lutz et al., 2016; Ismail and Bogacki, 2018; Hasson et al., 2019; Ismail et al., 2020).

In order to allow for a more process-based estimate of the *DDF*,  present study attempts to quantify *[handwritten note: Need 'the' here]* the contribution of each energy balance component to melt and subsequently to the overall *DDF*. *[handwritten note: snow]*

[revised manuscript text omitted]

(c)

[Figure]

*[handwritten note: on same line as panel A + B.]*

*[handwritten note: + QUALITY → EXPORT AS VECTOR OR HIGH-RES RASTER.]*

Figure 5 Variation of solar radiation based DDF$_S$ for a degree-day value of 1°C d for (a) different latitudes under constant snow albedo and clearness index; (b) snow albedos under constant latitude and clearness index; (c) different clearness indices under constant latitude and snow albedo – The latitude = 48° corresponds to the location of Brunnenkopfhütte test site.

**4.2 Longwave radiation component - DDF$_L$**

The net longwave energy flux $Q_L$ is calculated using eq. (21)(21), in which the outgoing radiation from the snowpack can be assumed as constant. Thus, the contribution of longwave radiation component DDF$_L$ is mainly dependent on air temperature and the emissivity of the atmosphere, in particular cloudiness conditions. Figure 66 and Table 2Table 2 present the DDF$_L$ as a function of degree-days $T_{DD}$, which are equivalent to the average daily air temperature, and cloudiness. For a wide range of degree-days, especially in conjunction with low cloudiness, the outgoing longwave energy flux is higher than the incoming, resulting in a theoretically negative degree-day factor that will reduce the total $DDF$. This means that the DDF$_L$ component under clear-sky conditions usually is rather contributing to a cooling of the snowpack than to melting. Under overcast conditions, the DDF$_L$ is relatively constant around 1 mm °C$^{-1}$ d$^{-1}$ with a maximum value of 1.3 mm °C$^{-1}$ d$^{-1}$ at $T_{DD}$ = 20 °C d. Although this contribution to the total $DDF$ is small compared to the shortwave radiation component DDF$_S$, it can be of importance at the onset of snowmelt in early spring, when the solar radiation is still low and the albedo of fresh snow is high.

Table 2 Longwave radiation component (DDF$_L$) [mm °C$^{-1}$ d$^{-1}$] for selected cloudiness [%] and degree-days [°C d]

| DDF$_L$ |
| --- |

*[handwritten note: SEEMS REDUNDANT WITH FIG. 6 ⟹ SUGGEST MOVING TO SUPPL. MAT, WILL ALSO HELP REDUCING THE (EXCESSIVE) LENGTH OF MANUSCRIPT.]*

[revised manuscript text omitted]

---

## Author Response (AR2)

**Estimating degree-day factors of snow based on energy flux components**

**Editor: Harry Zekollari (Comments and Responses)**

**Comments to the author**:

Dear Muhammad Fraz Ismail and colleagues,

Many thanks for having addressed all the comments by the four reviewers in great detail and for having updated your manuscript accordingly. Originally, I had anticipated to send out the manuscript for a second round of reviews but given your very detailed and adequate answering of all issues raised, I do not consider this to be necessary at this stage. Instead, I have re-read your manuscript entirely again, and provided a (relatively long) list of comments to be addressed in the PDF directly. Most of these are minor (including many grammar suggestions) and will require very little work, while a few comments are slightly more substantial and will require to have some extra info in the manuscript (but normally not requiring additional/new analyses to be performed).

Additionally, the figures require some work (see comments for every figure in the pdf directly), and there is room to shorten the overall length of the manuscript (as pointed out by some of the reviewers): I have indicated parts of the manuscript that are quite long (/ too long?), and/or not directly relevant for your story and that could therefore be removed or reduced (including some tables to be moved to suppl. mat.). Finally, while the topic you are addressing is timely, you refer in many cases to (only) (very old) studies: it would be good to have some more recent references in some places. The timeliness of your study is also clear from the recent debate that has (re)appeared on having constant (stationary) degree-day factors over time when modelling snow and ice melt. The debate strongly relates to whether degree-day approaches are capable of capturing non-linearities in melt, which may be important in a changing climate where the relationship between degree-days and melt is likely to change.

More specifically, I suggest you to have a look at the recent works by Bolibar et al. (2022, Nature Communications), Vincent and Thibert (2022, The Cryosphere Discussions), and Matthews et al. (preprint, https://www.researchsquare.com/article/rs-2166876/v1), and to potentially include these in your discussion and how your work fits in this existing 'debate'.

Thanks a lot for your work, and I look forward to seeing your updated manuscript.

Best regards,

Harry

Dear Editor,

Thank you very much for your constructive comments and suggestions to improve the manuscript. We are very grateful for the comprehensive manuscript summary and acknowledging our contribution. We have now updated all the figures as tables as per your comments and suggestions. We have also put all the necessary tables and figures in the supplementary materials. Some of the paragraphs are removed/reduced as per your recommendations. If we intend to retain any paragraph, then we also provide the reason for retaining it. We thank you very much for providing us with the updated recent references. We have included these updated references in the revised manuscript.

Based on your comprehensive comments and suggestion, we have made numerous changes in the revised version of our manuscript. Below, we repeat each of your comment and our reply to them one by one. All responses are in blue font for clarity of reading.

Muhammad Fraz Ismail

On behalf of all the authors

**DETAILED POINTS**

**Title:** "Of snow" (In general, need to specify this. Also, in some locations in text. Otherwise, could also be ice (+for outsiders is not given that they know this typically applies to snow and ice).

We have now changed the title of the manuscript to *Estimating degree-day factors **of snow** based on energy flux components* as per your recommendations.

**Line 55**: Update references!

We have now updated the references in the manuscript as per your comments and suggestions. A list of all new references has been provided at the end of this document.

**Line 99**: Quite a lot on this recently. Would be good to go a bit more in detail into this. See also general comment and work by Matthews et al. (under review), Bolibar et al. (2022), and Vincent and Thibert (2022).

We have now provided these references in the literature review as well as in the discussion section.

**Table 1**: Make sure this is on page when resubmitting

Table 1 is now on page. Kindly refer to the manuscript version without track changes.

**Figure 3**: Test to small SWE (mm w.e.) vs. [] I don't get this comment Axes too small Not sharp. Export/save as vector (e.g. PDF, SVG) instead of raster. Of higher res raster.

All the figures have been updated and all your comments have been addressed. We have now provided a separate PDF document with only high-resolution figures.

**Line 202**: Maybe remove?

"Unit area column" was mentioned according to a comment of RC4, but we appreciate using the original formulation as follows.

The energy flux available for snowmelt $Q_M$ can be calculated from the balance of energy fluxes entering or leaving the snowpack and the change in the internal energy stored in the snowpack $\Delta Q$.

**Line 275 – 277**: Then, probably best to remove entire part here (see next comment)

We would like to keep this paragraph in order to mention approaches that can be used in situations where only temperature data is available.

**Line 284 – 285**: Formula 14 & 15: Why two? What is difference? Specify or only give one.

In eq. (14) and (16) only the parameter $a$ of eq. (11) is a function of altitude $z$, while in eq. (15) and (17) also the parameter $b$ is dependent on $z$.

**Line 300**: Why? Would be good to give at least a hint.

We have rephrased the paragraph as follows, hoping it becomes clearer:

… It should be noted, that although $Kz$ is higher for overcast than for clear sky conditions, the absolute increase of the clearness index $K_T$ with altitude is higher under clear sky conditions because of the higher base value $K_{T_0}$. For example, at z = 2000 m a.s.l. model Jin (b) (eq. **Error! Reference source not found.**) has a clearness altitude factor $K_z$ of 1.27 for overcast and 1.10 for clear sky conditions. However, when multiplying by the respective clearness factors at sea level $K_{T_0}$ of 0.15 and 0.72, the resulting clearness indices $K_T$ at z = 2000 m a.s.l. increase by 0.04 under overcast and 0.07 under clear sky conditions to an absolute value of 0.19 and 0.79 respectively.

**Figure 4**: Increase font size! Figure will be reduced in size -> will be hard to read What is the difference between Jin (a) and Jin (b)? see also comment on this earlier. Remove all unused space in graphic Replace [] with ().

All the figures have been updated as per your comments and suggestions.

**Caption Figure 4**: What does this mean? Applicable to what?

We have now rephrased the caption as follows.

Clearness altitude factor $Kz$ for latitude 45° and different altitudes, based on different models presented in equations (14 – 17, i.e. Jin (a), Jin (b), Rensheng, and Liu).

**Line 310 – 311**: Twice the name. Don't have more recent reference?

These are two different papers. We have also added a more recent reference i.e., Amaral et al., (2017).

**Line 312**: Really? So old?

The basic exponential approach has not changed since then. There are more recent approaches to obtain albedo from satellite data, but we don't want to go into this.

**Line 358**: Screen Level: What is this?

The level at which the instruments take the measurements.

**Line 372 – 378**: Not needed here I would say. Too detailed. Focus on methodology that matters for your work!

This was an important comment of **RC1**. As we demonstrate effects of altitude (pressure), temperature and humidity on sensible and latent heat later, we think respective equations should be given.

**Line 380 – 382**: Does not seem to be relevant for your story here.

Example values are given to show the magnitude of density change.

**Line 388 – 392**: Possibly remove?

We have now removed the paragraph.

**Section 3.2.5**: So no need for this section. Be more compact, focus on the essence.

As suggested, we have deleted the section.

**Line 391**: How is this related to other half-year (which you do not show from my understanding?)

The curve for the other half of the year will be mirror image. That is why we only show half the part.

**Figure 5**: Make more compact, all 3 on one line, increase font size, because will be (much) smaller when typesetted + quality -> export as vector of high-res raster

All the figures have been updated as per your comments and suggestions.

**Table 2**: Seems redundant with fig. 6 -> suggest moving to suppl. Mat, will also help reducing the (excessive) length of manuscript.

We have now moved this table to the supplementary materials.

**Figure 6**: Legend not color blinding proof (e.g., Green + red) add info (also) next to respective line, Larger font size

All the figures have been updated as per your comments and suggestions.

**Line 574 – 579**: Suggest to remove. Aside-track from main story and not part of your results. Eventually to part of the discussion.

We have now removed the paragraph.

**Table 3** Merge with **table 4**

We have now deleted Table 3, and Table 4 has been moved to the Supplementary Materials because Figure 07 has been drawn from Table 4.

**Figure 7**: Legend not colorblind proof. Add info in figure also Remove blank space Larger font size Replace [] with ()

All the figures have been updated as per your comments and suggestions.

**Table 5**: To suppl. Mat. Info in Figure 8

Table 5 has been moved to the Supplementary Materials.

**Figure 8**: Remove blank space larger legend and not colourblind proof See remarks for fig. 6

All the figures have been updated as per your comments and suggestions.

**Figure 9**: Font size larger Reduce hight by about 50%

All the figures have been updated as per your comments and suggestions.

**Line 684**: Linked to cloudiness? Maybe give hint here.

We have added the hint to the relevant figure (see Figure 4).

**Line 695**: Over which altitude range?

We have now provided the altitude ranges from respective references. The updated paragraph reads as follows:

For example, in the Nepalese Himalayan region, seasonal-average *DDF* increases from 7.7 to 11.6 mm d$^{-1}$ °C$^{-1}$ with respect to altitude ranging 4900 to 5300 m a.s.l. (Kayastha et al., 2000) whereas Kayastha and Kayastha, (2020) found that the model-calibrated range of the *DDF* in central Himalayan basin varies between 7.0 – 9.0 mm d$^{-1}$ °C$^{-1}$ over an approximate altitude range of 4000 – 8000 m a.s.l.

**Line 705**: Include albedo info in figure!

We have included albedo in the figure.

**Line 734**: A bit vague when formulated as such. -> can you be more specific? The reason?

We have reformulated the paragraph as follows:

Under overcast conditions, DDF$_L$ is neutral or slightly positive while the DDF$_S$ component decreases because degree-days are rising faster than input from solar radiation, which implies that sky conditions (i.e., overcast, and clear sky) are more decisive for an estimate of the *DDF* than the day of the year.

**Line 742**: breaking in (What is this?)

The English terminology was suggested by **RC1**. Meaning a sudden and fierce change of weather conditions.

**Line 744 – 749**: Too detailed and not crucial in your story it seems. Suggest removing

We have now removed it as per your suggestion.

**Line 803**: Where is this shown (SNOWPACK results)? Did you do this? Not clear + provide info (e.g. in suppl. Mat).

We have now provided the graphs/figures in the supplementary materials section S2.

**Figure 11**: Data source and period in caption not in figure. How are these new snow events defined? When is it new?

All the figures have been updated as per your comments and suggestions. New snow events are defined with a threshold of precipitation $\geq$ 5mm d$^{-1}$.

**Line 818**: Why for this region and not for your study site? Would be good to explain.

The following para has been added.

In the current example, the Upper Jhelum catchment is discussed because of elevation zone wise data availability in comparison to test site where only point data is available (for more detail see Sec. 2).

**Line 823**: Where does future climate come from? Why RCP´s and no SSPS? Need more info here on how this is calculated Ok given later (see last comment on this page) can you provide this here?

The following para has been added.

Figure 12 shows the overall picture that how the *DDF* for snow will change over time and under climate change (i.e., Present, RCP2.6, RCP8.5 (for *DDF* estimates under climate change, see Sec. 5.2.3)).

**Line 830**: You (almost) never mention snow for DDF, so confusing when you do here. Suggest removing.

As per your suggestion, we have now modified the manuscript title as well as mentioned several times in the manuscript. We hope that now it is clear.

**Line 855**: Here could include studies Bolibar et al (2022) Vincent and Thibert (2022) + Matthews et al for discussion

We have included the necessary references in the revised manuscript.

**Figure 12**: Remove blank space, Legend and font larger, Quality : vector

We have retained the x-axis from February to August in all facets because we want to show how the DDFs vary over time and under climate change. In our opinion it would be nice to show the same time frame for all the facets. We have also incorporated your comments about legend size and picture quality.

**Line 869**: Most of which I suggest moving to suppl. Mat. Given limited added value compared to figures.

As per your suggestions, we have moved the necessary tables in the supplementary materials.

**Line 886: Conclusion:** Not sure I understand. How can DDF be moderate?

We have reformulated the sentence as follows:

Therefore, total *DDF* value is not very high and variations due to other factors are usually limited, apart from exceptional rainstorm events, for which energy balance models are the more suitable approach.

**List of new References:**

- Amaral, T., Wake, C. P., Dibb, J. E., Burakowski, E. A., and Stampone, M.: A simple model of snow albedo decay using observations from the Community Collaborative Rain, Hail, and Snow-Albedo (CoCoRaHS-Albedo) Network, J. Glaciol., 63, 877–887, https://doi.org/10.1017/jog.2017.54, 2017.

- Badescu, V. (Ed.): Modeling solar radiation at the earth's surface: recent advances, Springer, Berlin, 517 pp., 2008.

- Bolibar, J., Rabatel, A., Gouttevin, I., Zekollari, H., and Galiez, C.: Nonlinear sensitivity of glacier mass balance to future climate change unveiled by deep learning, Nat Commun, 13, 409, https://doi.org/10.1038/s41467-022-28033-0, 2022.

- Huss, M. and Hock, R.: Global-scale hydrological response to future glacier mass loss, Nature Clim Change, 8, 135–140, https://doi.org/10.1038/s41558-017-0049-x, 2018.

- Immerzeel, W. W., Lutz, A. F., Andrade, M., Bahl, A., Biemans, H., Bolch, T., Hyde, S., Brumby, S., Davies, B. J., Elmore, A. C., Emmer, A., Feng, M., Fernández, A., Haritashya, U., Kargel, J. S., Koppes, M., Kraaijenbrink, P. D. A., Kulkarni, A. V., Mayewski, P. A., Nepal, S., Pacheco, P., Painter, T. H., Pellicciotti, F., Rajaram, H., Rupper, S., Sinisalo, A., Shrestha, A. B., Viviroli, D., Wada, Y., Xiao, C., Yao, T., and Baillie, J. E. M.: Importance and vulnerability of the world's water towers, Nature, 577, 364–369, https://doi.org/10.1038/s41586-019-1822-y, 2020.

- Muhammad, S., Tian, L., Ali, S., Latif, Y., Wazir, M. A., Goheer, M. A., Saifullah, M., Hussain, I., and Shiyin, L.: Thin debris layers do not enhance melting of the Karakoram glaciers, Science of The Total Environment, 746, 141119, https://doi.org/10.1016/j.scitotenv.2020.141119, 2020.

- Oerlemans, J.: Glaciers and climate change, A.A. Balkema Publishers, Lisse ; Exton, (PA), 148 pp., 2001.

- Swinbank, W. C.: Long-wave radiation from clear skies, Q.J Royal Met. Soc., 89, 339–348, https://doi.org/10.1002/qj.49708938105, 1963.

- Vincent, C. and Thibert, E.: Brief communication: Nonlinear sensitivity of glacier-mass balance attested by temperature-index models, Glaciers/Alpine Glaciers, https://doi.org/10.5194/tc-2022-210, 2022.

- Yang, K. and Koike, T.: A general model to estimate hourly and daily solar radiation for hydrological studies: GENERAL SOLAR RADIATION, Water Resour. Res., 41, https://doi.org/10.1029/2005WR003976, 2005.

[Figure]

Brunnenkopfhütte 1602 m a.s.l.

**Legend**

● Brunnenkopfhütte

—— Contours - 50m

—— Streams

▢ Dreisäulerbach catchment

**DEM** (m a.s.l.)

High : 1766

Low : 946

0    0.25    0.5    1 Kilometers

N

Source: 5m DEM (EPSG:25832) Geobasisdaten, Bayerische Vermessungsverwaltung

1650  1700  1750  1600  1550  1500  1600  1600  1450  1400  1350  1300  1250  1200  1150  1100  1050  1000  1300  950

[Figure]

**List of sensors**

1. Humidity Sensor
2. Wind wane
3. Solar Radiation (incoming)
4. Temperature sensor
5. Snow depth sensor
6. Snow pack analyser
7. Snow scale
8. Data logger

Solar Radiation (Not shown)
Pluvio (Not shown)
Pressure sensor (Not visiable)